# Uncovering Intrinsic Capabilities: A Paradigm for Data Curation in Vision-Language Models

## Abstract

Large vision–language models (VLMs) achieve strong benchmark performance, but controlling their behavior through instruction tuning remains difficult. Reducing the budget of instruction tuning dataset often causes regressions, as heuristic strategies treat models as black boxes and overlook the latent capabilities that govern learning. We introduce Capability-Attributed Data Curation (CADC), a framework that shifts curation from task-specific heuristics to intrinsic capability analysis. CADC discovers intrinsic capabilities in an unsupervised manner from gradient-based learning trajectories, attributes training data to these capabilities via influence estimation, and curates capability-aware curricula through balanced selection and staged sequencing. This transforms black-box instruction tuning into a controllable, capability-driven process. With as little as 5% of the original data, CADC surpasses full-data training on multimodal benchmarks. These results validate intrinsic capabilities as the fundamental building blocks of model learning and establish CADC as a principle paradigm for instruction data curation.

## 1 Introduction

Instruction tuning is widely adopted to fine-tune large vision-language models (VLMs) (Dai et al., 2023), adapting them to a wide range of human-centric downstream tasks (Zhang et al., 2024). Consequently, various thematically diverse datasets are subtly curated inducing powerful generalization ability with a small fraction of instruction data (Lee et al., 2024; Zhou et al., 2023).

A central challenge to instruction tuning is how to best utilize these curated datasets. The straightforward attempts are made towards choosing the most similar out of domain data points to the in-domain data points (Pruthi et al., 2020; Liu et al., 2024c). Further to these data-feature-centric approaches, Xia et al. (2024) proposed to directly minimize the training loss in representing the targeted tasks, modeled as LLM **capability** (e.g., reasoning skill), instead of prioritizing the importance of data feature similarities.

Despite the paradigm shift from a *data-centric* to a *capability-centric* optimization framework, the capability could not be coarsely modeled as a gradient-based learning trajectory between training data points associated with a specific task (Xia et al., 2024; Wu et al., 2024). On the contrary, even a simple real-world task involves the complementarity of multiple **intrinsic capabilities**, i.e., the latent capabilities through which different training data points are mapping to accomplish a single task. For example, analyzing a chemical reaction diagram and explaining its mechanism requires structural grounding (to identify molecular structures and their relationships), perceptual recognition (to identify chemical entities and symbolic elements), and symbolic reasoning (to deduce reaction pathways). If the instruction data are optimized to disproportionately reinforce reasoning capability while neglecting the enhancement of recognition and grounding capabilities, the model would inevitably exhibit worse performance (Chen et al., 2025; Zhong et al., 2025). As for intrinsic capabilities, we have two empirical observations. First, we found that many curated instruction datasets, although they appear to be highly diverse, usually reside in the same low-dimensional manifold of intrinsic model capabilities. Second, even a single real-world task may involve multiple intrinsic capabilities to govern the instruction data points.

To discover multiple intrinsic capabilities and balance the contribution of each intrinsic capability, we propose Capability-Attributed Data Curation (CADC), a framework that shifts data curation from extrinsic task heuristics (Xia et al., 2024; Wu et al., 2024) to intrinsic capability analysis, as illustrated in Figure 1. CADC first discovers intrinsic capabilities in an unsupervised manner from gradient-based learning trajectories, then attributes training samples to these capabilities through influence estimation, and finally curates capability-aware subsets via balanced selection and curriculum sequencing. By aligning data with the capabilities that the model actually acquires, CADC transforms instruction tuning into a controllable process. Notably, CADC not only provides a structured view of model learning, but also achieves state-of-the-art efficiency: capability-aware selection enables small curated subsets to match full dataset performance, and sequencing them along the natural learning progression extends efficiency further, allowing as little as 5% of the original data to surpass the performance of training on 100%.

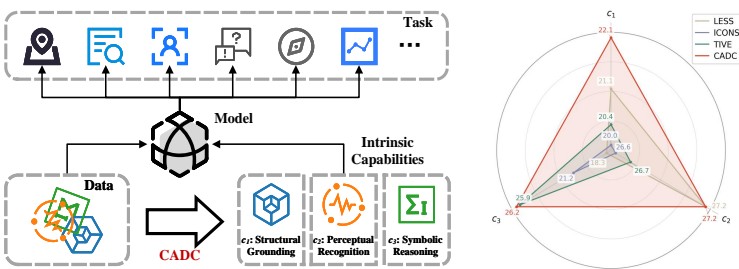

Figure 1: Motivation and capability analysis of CADC. Left: CADC disentangles mixed training data into groups aligned with intrinsic model capabilities and allocates them in a principled manner to support downstream tasks. Right: SmolVLM capability performance across $c_1$, $c_2$, and $c_3$, showing that CADC improves the model's capabilities in a balanced manner.

Our contributions are threefold:

- **Intrinsic capability discovery.** We design an unsupervised module to discover the latent capabilities of a VLM directly from its learning dynamics.
- **Capability-attributed mapping.** We design an influence-based attribution module to quantify how a data point contributes to specific capabilities and the widely adopted adamW algorithm is reformulated as the new VLM optimizer.
- **Capability-aware curation.** We optimize data curation process to balance the learning dynamics of multiple capabilities via the designed curriculum learning module.

## 2 PRELIMINARIES

To select less instruction data with minimal performance loss, we formalize the problem in terms of a model's **intrinsic capabilities** and identify three key challenges. We first define what intrinsic capabilities are and how to uncover them (Challenge 2.1). We then consider how to attribute these capabilities to the data (Challenge 2.2). Finally, we address how to modulate multiple capabilities during training (Challenge 2.3).

Recall that, interpreting a scientific diagram might require a combination of perceptual recognition capability $c_{pr}$ (to identify visual elements) and symbolic reasoning capability $c_{sr}$ (to infer relationships among those elements). Formally, an intrinsic capability can be defined as a latent skill such that performance on any task can be factorized into contributions from one or more of these capabilities (see Appendix A for details). However, there is no direct supervision for $\mathcal{C}$, the granularity of each capability is ambiguous, and different decompositions of $\mathcal{C}$ could explain the same observed behavior. A principled approach is required to discover the true underlying skills.

**Challenge 2.1** (Intrinsic Capability Identification). Given a model with parameters $\theta$ and an unknown set of latent intrinsic capabilities $\mathcal{C} = \{c_1, \ldots, c_K\}$ underpinning its behavior, where $K \in \mathbb{N}$ denotes the number of capabilities, infer a minimal coherent decomposition of the model's skills into distinct capabilities using only observable signals (e.g., training dynamics, performance patterns, or input-output behavior).

Given a set of intrinsic capabilities $\mathcal{C}$, the model acquires its intrinsic capabilities through data-driven learning, thus training data become a natural lever for control. By strategically manipulating the composition or presentation of the training data, we may steer the activation of particular capabilities (see Appendix B.1 for the derivation). However, the relationship between training data $z$, target data $z'$, and intrinsic capability $\mathcal{C}$ remains unclear, modeled as below research challenge.

**Challenge 2.2** (Attributing Intrinsic Capabilities to Data). Given the set of samples $\mathcal{D}$ (with their influence trajectories) and a set of intrinsic capabilities $\mathcal{C} = \{c_1, \ldots, c_K\}$, learn a mapping

$$\mathcal{A} : \mathcal{D} \to 2^{\mathcal{C}}, \tag{1}$$

that assigns each sample $z$ to a subset of capabilities $\mathcal{A}(z) \subseteq \mathcal{C}$ that it influences most significantly.

To balance the learning dynamics for multiple intrinsic model capabilities, the data curation process should principally guarantee that no individual capability dominates the rest capabilities. We approach this issue through *self-influence* of a data point. We define the self-influence of a training sample $z$ as the cumulative magnitude of its own gradient over training, written as $\mathrm{Inf}^{\mathrm{Self}}(z; M) \triangleq \sum_{i=1}^{M} \bar{\eta}_i \langle \nabla \ell(z, \boldsymbol{\theta}_i), \nabla \ell(z, \boldsymbol{\theta}_i) \rangle$ (see Appendix B.3 for details). Inspired by human curriculum learning (Wang et al., 2022), we assume that the intrinsic capabilities of a model should be acquired in a staged manner: fundamental skills first, then more complex skills built on top. This requires the designed mechanism could orchestrate when each capability is introduced during training and how to balance multiple intrinsic capabilities becomes our last research challenge.

**Challenge 2.3** (Modulating Intrinsic Capabilities via Data). Given a set of intrinsic capabilities $\mathcal{C} = \{c_1, \ldots, c_K\}$ and a training dataset $\mathcal{D}$ (where each sample $z \in \mathcal{D}$ has been attributed to one or more capabilities), design a training data curation strategy to satisfy two requirements: each capability $c_k$ obtains a sufficient training signal, and the capabilities are introduced in a purposeful staged order ($c_{i_1} \prec c_{i_2} \prec \cdots \prec c_{i_K}$).

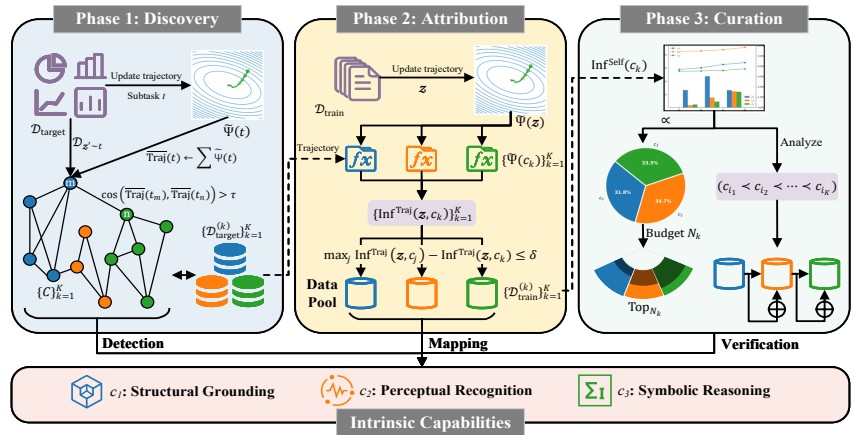

Figure 2: Overview of the CADC pipeline. The framework operates in three phases: (1) **Discovery** identifies intrinsic capabilities by clustering gradient-based learning trajectories of target data; (2) **Attribution** maps training samples to these capabilities through trajectory influence analysis, forming capability-specific data pools; (3) **Curation** leverages self-influence signals to allocate budgets and sequence data, enabling capability-aware curricula. The three discovered capabilities—structural grounding ($c_1$), perceptual recognition ($c_2$), and symbolic reasoning ($c_3$)—serve as the foundation for balanced and interpretable data curation.

## 3 METHODOLOGY: CAPABILITY-ATTRIBUTED DATA CURATION

The proposed *Capability-Attributed Data Curation (CADC)* consists of three main phases. The first phase is **unsupervised discovery of intrinsic capabilities** from the model's own learning dynamics, which addresses Challenge 2.1. The second phase involves conducting **capability attribution** to data, targeting Challenge 2.2. The third phase is **data curation for curriculum learning**, which aims to foster balanced and staged training of capabilities and addresses Challenge 2.3. Details are depicted in Figure 2 and techniques for each component are illustrated in the following subsections.

## 3.1 Unsupervised Discovery of Intrinsic Capabilities

To address Challenge 2.1, we propose an unsupervised, data-driven approach to discover a model's intrinsic capabilities directly from its learning dynamics. The key idea is to observe how the model learns on a broad set of validation tasks and identify clusters of tasks that induce similar learning behavior — each such cluster can be interpreted as an intrinsic capability.

Concretely, we begin with a comprehensive validation set containing a diverse collection of subtasks that span various domains (in our experiments, we use MMT-Bench, a multimodal multi-task benchmark, to approximate the range of human vision-language tasks). During model training, we periodically record the parameter update trajectory for each validation sample. As AdamW optimizer is widely used for VLM fine-tuning, CADC adopts the update signal $\mathcal{U}_{\text{AdamW}}\big(\nabla \ell(\cdot; \cdot), \cdot\big) = \tilde{\Psi}(\cdot; \cdot)$, as defined in Appendix B.2. For a given validation sample $z'$ at checkpoint $\theta_i$, we calculate its AdamW (Loshchilov & Hutter, 2019) update vector $\tilde{\Psi}(z'; \theta_i)$ using Eq. 13. Next, we aggregate these trajectories by subtask. Let $\mathcal{T}$ denote the set of target subtasks in the validation set. For each subtask $t \in \mathcal{T}$, we average the compressed trajectories of its samples:

$$\overline{\text{Traj}}(t) = \sum_{i=1}^{M} \bar{\eta}_i \tilde{\Psi}(t; \theta_i) = \sum_{i=1}^{M} \bar{\eta}_i \mathbb{E}_{z' \sim t}[\tilde{\Psi}(z'; \theta_i)]. \tag{2}$$

This vector $\overline{\text{Traj}}(t)$ summarizes how the model learns the subtask $t$ during training.

We then perform community detection on these subtask trajectories to uncover intrinsic capabilities. We construct an undirected task-similarity graph $\mathcal{G} = (V, E)$ where each node $v \in V$ represents a target subtask, and an edge connects two subtasks $t_m$ and $t_n$ if the cosine similarity between their vectors $\overline{\text{Traj}}(t_m)$ and $\overline{\text{Traj}}(t_n)$ exceeds a threshold $\tau$. This graph connects tasks that behave similarly from the model learning perspective. We apply the Leiden community detection algorithm (Traag et al., 2018) to partition $\mathcal{G}$ into $K$ disjoint clusters (communities) of subtasks: $\{C_1, \ldots, C_K\}$.

Each cluster $C_k$ is a set of subtasks that induce similar learning dynamics in the model; we interpret this cluster as an intrinsic capability $c_k$. Thus, each discovered capability $c_k$ comes with its representative subtasks (the cluster $C_k$) and their associated validation data, which we denote $\mathcal{D}_{\text{target}}^{(k)}$. By this process, we obtain a set of intrinsic capabilities $\mathcal{C} = \{c_1, \ldots, c_K\}$ in an unsupervised manner, thereby addressing Challenge 2.1. Importantly, this discovery does not assume any priori task taxonomy but instead lets the model's own dynamics reveal the capability structure.

## 3.2 Capability Attribution

Having identified intrinsic capabilities $\mathcal{C}$ and mapped the target data to these capabilities, we further tackle Challenge 2.2: mapping the training data to the discovered capabilities. For each training data point $z \in \mathcal{D}_{\text{train}}$, we measure how much $z$ influences each capability $c_k$ by quantifying the influence (Pruthi et al., 2020) of $z$ on the representative validation set $\mathcal{D}_{\text{target}}^{(k)}$:

$$\text{Inf}_{\text{AdamW}}^{\text{Traj}}(z, c_k) = \mathbb{E}_{z' \sim c_k}\Big[\sum_{i=1}^{M} \bar{\eta}_i \cdot \cos(\tilde{\Psi}(z; \theta_i), \tilde{\Psi}(z'; \theta_i))\Big] = \mathbb{E}_{z' \in \mathcal{D}_{\text{target}}^{(k)}}\Big[\sum_{i=1}^{M} \bar{\eta}_i \cdot \frac{\langle \tilde{\Psi}(z; \theta_i), \tilde{\Psi}(z'; \theta_i) \rangle}{\|\tilde{\Psi}(z; \theta_i)\| \|\tilde{\Psi}(z'; \theta_i)\|}\Big], \tag{3}$$

A high value of $\text{Inf}^{\text{Traj}}(z, c_k)$ means that training in sample $z$ updates the model in directions that strongly align with how the model learns capability $c_k$.

Using these influence scores, a naive approach would assign each training sample to the single capability $c_k$ with the highest $\text{Inf}^{\text{Traj}}(z, c_k)$. However, this winner-takes-all assignment can be brittle and overly restrictive, since many training samples are versatile — they simultaneously contribute to multiple capabilities. To account for this, we adopt a soft and non-exclusive attribution with a tolerance threshold $\delta \geq 0$. We assign a training sample $z$ to the pool $\mathcal{D}_{\text{train}}^{(k)}$ of capability $c_k$ if the influence of $z$ on $c_k$ is within $\delta$ of its maximum influence across all capabilities:

$$\mathcal{D}_{\text{train}}^{(k)} = \{z \in \mathcal{D}_{\text{train}} \mid \max_{i=1,\ldots,K}\{\text{Inf}^{\text{Traj}}(z, c_i)\} - \text{Inf}^{\text{Traj}}(z, c_k) \leq \delta\}. \tag{4}$$

When $\delta = 0$, this reduces to a strict winner-take-all (each $z$ is assigned only to its top capability); a larger $\delta$ allows a sample to be shared among multiple capability pools if its top influence scores are nearly tied. This tolerance-based assignment acknowledges the uncertainty in the attribution.

The outcome of this step is the partitioning of the training dataset into $K$ capability-specific pools $\{\mathcal{D}_{\text{train}}^{(1)}, \ldots, \mathcal{D}_{\text{train}}^{(K)}\}$ (with potential overlap if $\delta > 0$). At this point, we know which training examples are most relevant for learning each intrinsic capability.

## 3.3 DATA CURATION FOR CURRICULUM LEARNING

Finally, we leverage the data-to-capability map to curate training data and design a curriculum addressing Challenge 2.3, considering two facets: (1) arrangement, selecting a balanced high-value subset, and (2) sequencing, determining the order to introduce capability-specific data.

### 3.3.1 CURRICULUM ARRANGEMENT

The curriculum arrangement focuses on selecting a subset of training data that is both capability-balanced and high-quality. Suppose that our goal is to choose $N$ training samples in total for fine-tuning. We proceed in two steps:

**Budget Allocation.** We quantify the "learning difficulty" of capability $c_k$ by self-influence:

$$\text{Inf}^{\text{Self}}(c_k) = \text{Inf}^{\text{Self}}(\mathcal{D}_{\text{train}}^{(k)}) = \mathbb{E}_{\boldsymbol{z} \in \mathcal{D}_{\text{train}}^{(k)}}\big[\text{Inf}^{\text{Self}}(\boldsymbol{z}; M)\big]. \tag{5}$$

A higher value means that the model struggles more with the data for the capability $c_k$. Allocate budget $N_k \propto \text{Inf}^{\text{Self}}(c_k)$ for each capability: $N_k = \frac{\text{Inf}^{\text{Self}}(c_k)}{\sum_{i=1}^{K} \text{Inf}^{\text{Self}}(c_i)} \times N$.

**Pool Sampling.** For each capability's pool $\mathcal{D}_{\text{train}}^{(k)}$, we rank its samples by their relevance to capability $c_k$. We use the trajectory influence score $\text{Inf}^{\text{Traj}}(\boldsymbol{z}, c_k)$ as a measure of how useful the sample $\boldsymbol{z}$ is to improve $c_k$. We select the first $N_k$ samples from $\mathcal{D}_{\text{train}}^{(k)}$ with the highest $\text{Inf}^{\text{Traj}}(\boldsymbol{z}, c_k)$.

This two-step arrangement yields a capability-balanced subset that highlights the most informative samples. By preventing dominant capabilities from overshadowing weaker ones and reducing redundancy, it fosters stable capability growth while alleviating overfitting and catastrophic forgetting.

### 3.3.2 CURRICULUM SEQUENCE

Curriculum sequencing determines the order in which the model is exposed to the capability-specific data pools. Instead of a random or simultaneous mix of all data, we hypothesize that aligning the training order with the natural learning progression will yield fewer conflicts and greater stability. To discover a suitable sequence, we analyze how each capability is learned in the training stages.

We track the self-influence (Bejan et al., 2023) curve for each capability. Specifically, for each capability $c_k$, we examine $\text{Inf}^{\text{Self}}(\mathcal{D}_{\text{train}}^{(k)}, i)$ – the average self-influence of its pool of stage $i$. If the self-influence of one capability rises sharply or starts high in the early stages, it suggests the model is learning that capability early, making it more foundational. In contrast, a capability that improves only later likely depends on the prior acquisition of other capabilities. By comparing these trends, we rank the capabilities to infer a plausible learning order $c_{i_1} \prec c_{i_2} \prec \cdots \prec c_{i_K}$. For example, we might observe that the model naturally focuses on perceptual recognition before it improves in symbolic reasoning, indicating that $c_{\text{pr}}$ should precede $c_{\text{sr}}$ in the curriculum.

Training is scheduled in phases aligned with the inferred capability order: the model first focuses on $c_{i_1}$, then on $c_{i_2}$, etc. To mitigate forgetting, each phase retains a small fraction of the earlier data rather than completely excluding it (Dong et al., 2024). This sequencing addresses Challenge 2.3 by reducing interference and competition for model capacity, fostering stable capability development.

In summary, CADC transforms curriculum design into a model-informed process by letting learning dynamics dictate *what* data to use and *when*. Guided by intrinsic capabilities, CADC shifts instruction data curation from ad-hoc mixing to a principled, capability-centric paradigm. The corresponding algorithm is depicted in the Appendix C.

## 4 EXPERIMENTS

We conduct experiments on **LLaVA-1.5 Mix665K** (Liu et al., 2024a) and **Vision-Flan** (Xu et al., 2023), and evaluate across diverse benchmarks including **LLaVA-Wild Bench** (Liu et al., 2023),

Table 1: Relative performance (%) of LLaVA under different data selection methods. Values are normalized to the performance of training on the full dataset (100%). *Data %* denotes the proportion of training data used. *Rel. Avg.* is the average over benchmarks. Names in parentheses denote the data selection model and its parameter size. ↑ indicates larger is better.

(a) Results with 15% training data. Comparative results taken from TIVE (Liu et al., 2024c).

| Method | Data % | SEED | SQA | | MMBench | | POPE↑ | Rel. Avg.↑ |
| | | Image↑ | Full↑ | Image↑ | EN↑ | CN↑ | | |
|---|---|---|---|---|---|---|---|---|
| Random | 15% | 93.6 | 100.6 | 102.4 | 96.1 | 93.5 | 97.7 | 97.3 |
| Length | 15% | 92.6 | 102.4 | 103.6 | 92.2 | 92.5 | 97.0 | 96.7 |
| Perplexity | 15% | 92.7 | 101.6 | 101.6 | 96.9 | 94.3 | 97.3 | 97.4 |
| GraNd | 15% | 94.3 | 102.9 | 102.4 | 97.8 | 93.1 | 96.0 | 97.8 |
| EL2N | 15% | 93.6 | 101.2 | 99.1 | 95.8 | 96.2 | 98.5 | 97.4 |
| TIVE (LLaVA-v1.5-7B) | 15% | 95.6 | 104.0 | **105.7** | 101.1 | 99.8 | 99.7 | 101.0 |
| | 5% | 95.1 | 106.3 | 103.9 | 100.3 | 108.5 | 99.4 | 102.2 |
| CADC (SmolVLM-256M) | 15% | **98.7** | **107.7** | 105.3 | **103.3** | 112.5 | 100.5 | **104.7** |
| | 20% | **98.7** | 106.6 | 104.0 | 102.6 | **113.2** | 100.9 | 104.3 |

(b) Results with 20% training data. Comparative results taken from ICONS (Wu et al., 2024).

| Method | Data % | SQA | MMBench | | POPE↑ | VQAv2↑ | LLaVA-W↑ | Rel. Avg.↑ |
| | | Image↑ | EN↑ | CN↑ | | | | |
|---|---|---|---|---|---|---|---|---|
| Random | 20% | 100.1 | 94.1 | 93.0 | 98.0 | 95.7 | 95.7 | 96.1 |
| CLIP-Score | 20% | 95.0 | 83.5 | 88.3 | 98.7 | 92.8 | 97.5 | 92.6 |
| EL2N | 20% | 95.8 | 80.5 | 80.5 | 97.6 | 96.3 | 95.6 | 91.0 |
| Perplexity | 20% | 95.2 | 78.7 | 77.8 | 95.6 | 95.8 | 100.6 | 90.6 |
| D2-Pruning | 20% | 101.3 | 99.4 | 97.8 | 99.2 | 92.3 | 94.1 | 97.3 |
| Self-Sup | 20% | 99.1 | 92.9 | 91.3 | 96.6 | 94.7 | 93.2 | 94.7 |
| Self-Filter | 20% | 89.8 | 73.8 | 76.9 | 97.0 | 93.2 | 95.6 | 87.7 |
| COINCIDE (TinyLLaVA-2B) | 20% | 101.2 | 95.5 | 92.5 | 99.7 | 96.7 | 99.1 | 97.4 |
| ICONS (LLaVA-v1.5-7B) | 20% | 103.5 | 95.5 | 94.7 | **101.3** | 96.5 | 97.3 | 98.1 |
| | 5% | 103.9 | 100.3 | 108.5 | 99.4 | 94.2 | **101.2** | 101.2 |
| CADC (SmolVLM-256M) | 15% | **105.3** | **103.3** | 112.5 | 100.5 | 100.4 | 97.6 | **103.3** |
| | 20% | 104.0 | 102.6 | **113.2** | 100.9 | **101.4** | 94.8 | 102.8 |

**VQAv2** (Goyal et al., 2017), **POPE** (Li et al., 2023b), **MM-Bench** (Liu et al., 2024b), **ScienceQA** (Lu et al., 2022), **SEED-Bench** (Li et al., 2023a), **RealWorldQA** (xAI, 2024), **HallusionBench** (Guan et al., 2024), **TextVQA** (Singh et al., 2019), **DocVQA** (Mathew et al., 2021), and **MMT-Bench** (Ying et al., 2024). Experiments are conducted with **SmolVLM** (Marafioti et al., 2025) and **LLaVA-v1.5** (Liu et al., 2024a). We compare against a broad set of baselines: **Random**, **Length**, **Perplexity** (Marion et al., 2023), **CLIP-Score** (Radford et al., 2021), **D2-Pruning** (Maharana et al., 2024), **EL2N** and **GraNd** (Paul et al., 2021), **Self-Sup** (Sorscher et al., 2022), **Self-Filter** (Chen et al., 2024), **LESS** (Xia et al., 2024), **TIVE** (Liu et al., 2024c), **COINCIDE** (Lee et al., 2024), and **ICONS** (Wu et al., 2024). [1] More details are provided in Appendices F and G. The following sections present the main results (§4.1), analysis (§4.2) and findings (§4.3).

## 4.1 MAIN RESULTS

The main experimental results are reported in Table 1 and Figure 3. Table 1 reports performance of data pruning using LLaVA-v1.5-7B and Figure 3a reports results in a unified experimental setting using SmolVLM-256M as training model. From these results, we have following observations.

**Data efficiency.** From this table, it is observed that CADC using less data consistently achieves better model performance than those baselines using full-data. In LLaVA-7B (Table 1), CADC with 5% of the data outperforms alternatives such as TIVE, COINCIDE, and ICONS, despite those methods relying on 15–20% budgets. In SmolVLM-256M (Figure 3a), CADC is also superior to the 100% baseline with only 5% data, reaching a relative average of 107.1%. Notably, CADC achieves the best results on almost all benchmarks, ranking second only on POPE and SQA. These results establish CADC as a highly efficient and robust paradigm for instruction data curation.

**Generalization across tasks and setups.** CADC delivers consistent and balanced improvements across tasks, as reflected in its best or second-best scores on all benchmarks. Not only avoids

---

[1] Abbreviations used throughout: Mix665K = LLaVA-1.5 Mix665K, LLaVA-W = LLaVA-Wild Bench, SQA = ScienceQA, SEED = SEED-Bench, RWQA = RealWorldQA, Hallusion = HallusionBench, Doc = DocVQA, MMT = MMT-Bench.

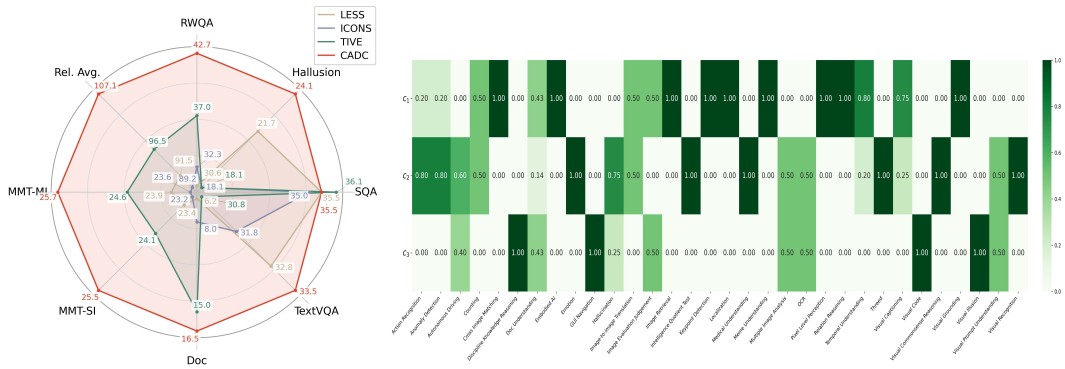

(a) Performance of SmolVLM under different data selection methods.

(b) Intrinsic capability discovery on MMT-Bench. Subtask distribution from 32 meta-tasks across three discovered capabilities.

Figure 3: Intrinsic capabilities discovered on MMT-Bench.

Table 2: Performance of SmolVLM models of different scales on Mix665K and Vision-Flan with 5% data. *CADC* denotes subsets selected with each model individually, and *CADC-T* denotes subsets selected using SmolVLM-256M's gradient store.

| | Dataset | Mix665K | | | Vision-Flan | | |
|---|---|---|---|---|---|---|---|
| Model | Data % | Random | CADC-T | CADC | Random | CADC-T | CADC |
| SmolVLM-256M | 5% | 96.4 | 107.1 | 107.1 | 82.6 | 87.7 | 87.7 |
| SmolVLM-500M | 5% | 65.1 | 91.3 | 96.0 | 68.4 | 72.4 | 78.8 |
| SmolVLM-2.2B | 5% | 84.1 | 95.4 | 89.0 | 75.8 | 79.9 | 92.7 |

regressions, it also shows clear advantages in challenging settings such as SEED and MMBench. This robustness holds across scales: CADC performs strongly with both LLaVA-7B and SmolVLM-256M, underscoring the generality of our capability-aware curation.

**Capability-aware coordination.** CADC disentangles intrinsic capabilities, alleviating conflicts inherent in conventional training. As shown in Figure 3a, it achieves large gains in benchmarks such as Hallusion and MMT, where heuristic and task-driven methods yield lower performance. By balancing complementary capabilities, CADC avoids the instability of approaches that overemphasize narrow signals, yielding more stable and reliable improvements across benchmarks.

**Validation of the intrinsic capability framework.** To summarize, the results Tables 1 and Figure 3a validate the effectiveness of the proposed CADC. First, CADC is competitive with the SOTA methods under heterogeneous conditions. Second, in a unified environment, CADC outperforms all baselines by a clear margin, validating that intrinsic capabilities are not only interpretable but also practically useful for driving data efficiency.

## 4.2 ANALYSIS

### 4.2.1 TRANSFERABILITY

We evaluate CADC transferability from three different perspectives. Table 2 reports the results for SmolVLM models of different scales (256M, 500M, 2.2B) on the Mix665K and on the Vision-Flan dataset. (1) **Model transferability.** CADC performs consistently well across model sizes, with capability-aware subsets outperforming random sampling in all variants of SmolVLM. (2) **Data transferability.** CADC-T, which uses subsets selected by SmolVLM-256M for larger models, achieves a performance comparable to model-specific curation, showing that small models can efficiently generate reusable subsets for larger ones. (3) **Training transferability.** On Vision-Flan, CADC outperforms random selection, confirming that its capability-driven strategy generalizes beyond Mix665K. These results establish CADC as a reusable curation framework that generalizes across models, transfers subsets across scales, and remains effective on diverse datasets.

Table 3: Ablation study of CADC across three aspects: (i) inclusion of key components (capability discovery, budget allocation, pool sampling, sequencing), (ii) curriculum sequencing orders, and (iii) budget allocations across capabilities. Reported values are relative to the full-data baseline (100%).

| Method | Component | | | | | Sequence | | Proportion | | | |
|---|---|---|---|---|---|---|---|---|---|---|---|
| | Capability | Budget | Pool | Sequence | Rel. Avg.↑ | Sequence | Rel. Avg.↑ | weight $c_1$ | weight $c_2$ | weight $c_3$ | Rel. Avg.↑ |
| **Random** | ✗ | ✗ | ✗ | ✗ | 96.4% | - | 96.4% | - | - | - | 96.4% |
| | | | | | | $(c_3 \prec c_2 \prec c_1)$ | 100.1% | 1.00 | 0.00 | 0.00 | 90.2% |
| | | | | | | $(c_3 \prec c_1 \prec c_2)$ | 100.6% | 0.00 | 1.00 | 0.00 | 75.4% |
| | ✗ | ✓ | ✓ | ✓ | 93.2% | $(c_3 \prec c_1 \prec c_2)$ | 100.6% | 0.00 | 0.00 | 1.00 | 92.9% |
| **CADC** | ✓ | ✗ | ✓ | ✓ | 104.0% | $(c_2 \prec c_3 \prec c_1)$ | 92.6% | 0.48 | 0.26 | 0.25 | **104.8%** |
| | ✓ | ✓ | ✗ | ✓ | 96.6% | $(c_2 \prec c_1 \prec c_3)$ | 93.9% | 0.24 | 0.52 | 0.25 | 97.6% |
| | ✓ | ✓ | ✓ | ✗ | 97.4% | $(c_1 \prec c_3 \prec c_2)$ | 104.2% | 0.24 | 0.26 | 0.50 | 102.2% |
| | ✓ | ✓ | ✓ | ✓ | **107.1%** | $(c_1 \prec c_2 \prec c_3)$ | **107.1%** | 0.32 | 0.35 | 0.33 | **107.1%** |

Table 4: Effect of the task-graph connectivity threshold $\tau$ on cluster count and model performance. *w/o Cluster* denotes no clustering, and *# Cluster* indicates the cluster count.

| Threshold | $\tau = 0$ | $\tau = 0.05$ | $\tau = 0.1$ | $\tau = 0.15$ | $\tau = 0.2$ | $\tau = 0.25$ | $\tau = 0.3$ | $\tau = 0.35$ | $\tau = 0.45$ | w/o Cluster |
|---|---|---|---|---|---|---|---|---|---|---|
| **# Cluster** | 3 | 3 | 3 | 3 | 3 | 3 | 4 | 7 | 12 | 1 |
| **Rel. Avg.↑** | 99.7% | 99.3% | 97.1% | 99.8% | **100.0%** | 99.7% | 96.7% | 97.6% | 97.1% | 95.8% |

### 4.2.2 ABLATION STUDY

Table 3 reports the results of ablation study from three perspectives: components, sequencing, and budget allocation. From this table, we have the following observations. (1) From a **component** perspective, removing any single module reduces performance, with the absence of capability discovery causing the largest drop, underscoring its central role in CADC (detailed component ablations are reported in Table 11). (2) From a **sequence** perspective, different curriculum orders yield consistently strong results, with the best results following the natural progression of $c_1 \prec c_2 \prec c_3$. This indicates that the CADC sequencing principle is well aligned with the model learning dynamics, while maintaining robustness among the alternatives. (3) From a **proportional** perspective, varying the sampling quotas across capabilities reveals that balanced or demand-aware allocations outperform extreme distributions. This shows that CADC not only controls training through principled allocation but also maintains stability under different weighting schemes.

### 4.2.3 CLUSTERING ROBUSTNESS

To verify the discovered intrinsic capabilities and assess the robustness of our capability-discovery method, we examine how the task-graph connectivity threshold $\tau$ affects the resulting cluster count and model performance (Table 4). To isolate clustering sensitivity, we set the tolerance threshold $\delta = 0$ and disable curriculum sequencing. As shown in Table 4, the clustering results are largely insensitive to $\tau$: across a wide range of values, the method consistently produces three clusters, indicating that the identified capabilities are intrinsic and stable attributes of the model.

To ensure that these capabilities are not artifacts of using MMT-Bench as the target dataset, we repeat the analysis with ICONS. As shown in Table 5, the ICONS target data are likewise partitioned into three clusters, supporting the objectivity of the discovered capabilities. Moreover, CADC achieves performance exceeding the full-data baseline using only 5% of the data, demonstrating the robustness of the proposed approach.

Table 5: Cluster counts and model performance on different target datasets, with performance reported relative to the full-data baseline.

| Method | Target Datasets | # Cluster | Rel. Avg.↑ |
|---|---|---|---|
| **Random** | - | - | 96.4% |
| **CADC** | MMT-Bench | 3 | **107.1%** |
| | ICONS | 3 | 102.8% |

### 4.2.4 COMPUTATIONAL COMPLEXITY

As summarized in Tables 6 and 7, CADC's cost is dominated by a one-time gradient feature computation step that, under our lightest snapshot configuration, scales linearly with data size ($|\mathcal{D}_{\text{train}}|$, $|\mathcal{D}_{\text{target}}|$) and snapshot count ($M$), while the subsequent data curation stage is comparatively neg-

Table 6: Computational complexity, wall-clock time (single-node hours) and storage cost per step. Steps run offline, so the cost is incurred only once. $m$ denotes the projected gradient dimension.

Table 7: Performance of CADC under different snapshot counts.

| | Gradient Features Computation | Data Curation |
|---|---|---|
| **Complexity** | $\mathcal{O}\Big((|\mathcal{D}_{\text{train}}| + |\mathcal{D}_{\text{target}}|) \cdot M\Big)$ | $\mathcal{O}\Big((|\mathcal{D}_{\text{train}}| \cdot |\mathcal{D}_{\text{target}}| + |\mathcal{T}|^2) \cdot m\Big)$ |
| **Wall-Clock** | 3.6 Hours | < 1 Min |
| **Storage** | 5.8 GB | - |

| | Rel. Avg.↑ |
|---|---|
| Random | 96.4% |
| $M = 1$ | 106.4% |
| $M = 4$ (default) | **107.1%** |

ligible. Both stages run entirely offline, so their cost can be amortized over multiple instruction-tuning runs and model variants. Moreover, even a single-snapshot configuration already attains performance close to our default multi-snapshot setting, indicating that choosing a smaller $M$ can significantly reduce offline cost with minimal impact on effectiveness.

Beyond the offline analysis above, we also report training wall-clock time in our Qwen2-VL experiments (Fig. 4). All Qwen2-VL backbones are fine-tuned either on 100% of Mix665K or on the same 5% subset obtained once by running CADC on the SmolVLM-256M selector. For larger backbones, training on the 5% CADC subset is substantially cheaper than training on the full data, while achieving comparable or better performance. Even counting the one-off selection time, "selection + 5% CADC training" is still cheaper than 100%-data training for large VL backbones, indicating that CADC remains compute-efficient in realistic large-model regimes.

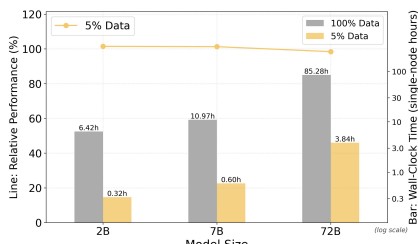

Figure 4: Relative performance and training wall-clock time of Qwen2-VL models (2B/7B/72B).

### 4.3 FINDINGS

In addition to the main results and abaltion study, we have following interesting findings which further verify the effectiveness of our approach.

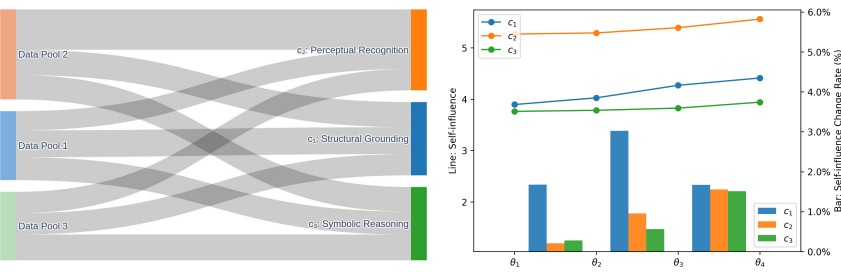

Figure 5: Influence of instruction training data. Left: Sankey diagram plots trajectory influence $\text{Inf}^{\text{Traj}}$ from the training pool $\mathcal{D}_{\text{train}}^{(k)}$ to capabilities $c_k$, with link thickness proportional to magnitude. Right: evolution of self-influence $\text{Inf}^{\text{Self}}$, where lines trace trends and bars show change rates.

**Finding 1: capability-label misalignment.** MMT-Bench groups its 162 subtasks into 32 meta-tasks, but CADC reveals that the model's learning behavior converges into only three intrinsic capabilities. **Structural Grounding** ($c_1$) refers to the ability to reason about spatial and structural relationships (e.g., *scene graph recognition*). **Perceptual Recognition** ($c_2$) denotes the ability to identify and classify objects, attributes, and scenes (e.g., *animal recognition*). **Symbolic Reasoning** ($c_3$) captures the ability to perform abstract, symbolic, and logical reasoning (e.g., *chart VQA*). The complete mapping of meta-tasks and subtasks to these capabilities are provided in Appendix D.

As shown in Figure 3b, the capability-based organization diverges from the benchmark-defined meta-tasks: subtasks within a single meta-task may be distributed across different capability clusters. For example, *existence hallucination* and *relation hallucination*, both labeled as *hallucination*

in MMT-Bench, are assigned to $c_2$ and $c_3$, respectively. This indicates that intrinsic capabilities more faithfully capture the latent structure of model learning than externally defined categories.

**Finding 2: data–capability alignment.** The left subfigure of Figure 5 shows that each training pool influences multiple intrinsic capabilities, and its dominant contribution aligns with a specific one. CADC disentangles these overlaps, organizing mixed signals into capability streams aligned with the model learning dynamics. This attribution has two advantages. First, it exposes hidden cross-capability effects that explain why task-based grouping often leads to interference. Second, it reorganizes training data into interpretable capability mappings, allowing balanced allocation and supporting curriculum sequencing. In this way, CADC transforms inherent overlaps into structured signals for controllable model training.

**Finding 3: curriculum signals from self-influence.** The right subfigure of Figure 5 plots the self-influence of each capability in the training stages. The resulting temporal profiles are clearly distinct, indicating that the disentangled capabilities function as distinct skills rather than clustering artifacts. Crucially, the rate of self-influence change provides a quantitative signal of demand: rising rates indicate greater marginal benefit from additional data attributed to that capability, while declining rates indicate saturation. A training curriculum, sequenced as $(c_1 \prec c_2 \prec c_3)$ based on these signals, consistently outperforms both random ordering and unsequenced selection (see Table 3). These results demonstrate that the intrinsic capability view is both interpretable and actionable. Self-influence also informs the allocation of sample budgets between capabilities (§3.3.1), with the corresponding sampling details for the training dataset provided in Appendix E.

## 5 RELATED WORK

We briefly review related works from two perspectives. For **data selection for model efficiency**, a central challenge to train vision–language models (VLMs) is the inefficiency and redundancy of instruction-tuning data (Zhou et al., 2023). Existing approaches tackled this issue using heuristics such as instruction length, perplexity (Marion et al., 2023), or embedding similarity (e.g., CLIP-Score (Radford et al., 2021)), while more advanced works exploited gradient signals to score or prune samples (e.g., GraNd Paul et al. (2021), EL2N (Paul et al., 2021)). Recently, influence-based methods have emerged, including LESS (Xia et al., 2024), ICONS (Wu et al., 2024), and TIVE (Liu et al., 2024c), which estimate sample utility via optimization dynamics, and COINCIDE (Lee et al., 2024), which emphasizes concept–skill diversity for better transfer. Although effective, these approaches remain task-driven, tying selection to external benchmarks or human-defined categories that may not reflect the latent structure of model learning (Zhou et al., 2025). CADC departs from this paradigm by uncovering and leveraging intrinsic capabilities directly from learning dynamics, enabling principled curation that align with how models actually acquire skills. For **Capability-aware data curation**, beyond selecting high-value data, several works highlight the importance of curriculum and diversity in training, yet most rely on ad hoc heuristics or task-oriented assumptions (Chrestien et al., 2023; Foglino et al., 2019; Chrestien et al., 2021; Lyu et al., 2025). Benchmarks such as MMT-Bench (Ying et al., 2024) further reveal that human task labels often conflate heterogeneous skills, suggesting the need for a more principled organizing principle. Our work introduces Capability-Attributed Data Curation (CADC), which reframes data management around intrinsic capabilities—latent skills discovered directly from learning dynamics. By mapping training samples to capabilities and balancing their growth through sequencing, CADC provides both interpretability and efficiency, enabling small curated subsets to achieve or surpass full-data performance.

## 6 CONCLUSION

We presented Capability-Attributed Data Curation (CADC), a framework that uncovers a model's intrinsic capabilities from its learning dynamics and leverages them for balanced selection and curriculum sequencing of training data. Experiments show that CADC not only reveals latent structures distinct from human-defined task labels but also delivers state-of-the-art data efficiency: with as little as 5% of the data, curated subsets can match or surpass full-data baselines. Furthermore, ablation and transfer experiments confirm its robustness in different settings and applicability across models and datasets. By aligning data curation with the capabilities a model actually learns, CADC establishes a principled and efficient paradigm for supervised fine-tuning.

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

## A  INTRINSIC CAPABILITY

**Definition A.1** (Intrinsic Capability). Consider a model with parameters $\boldsymbol{\theta}$ and a set of tasks $\mathcal{T}$. For any $t \in \mathcal{T}$, the performance $P(t; \boldsymbol{\theta})$ can be factorized as:

$$P(t; \boldsymbol{\theta}) = \Phi_t\Big(c_{k_1}, c_{k_2}, \ldots, c_{k_m}\Big), \quad c_{k_i} \subseteq \mathcal{C}, \tag{6}$$

where each $c_{k_i}$ is an intrinsic capability in the model's capability set $\mathcal{C} = c_1, \ldots, c_K$, and $\Phi_t$ is a task-specific composition function.

## B  INFLUENCE

### B.1  TRAJECTORY INFLUENCE

As demonstrated in LESS (Xia et al., 2024), the effect of a single training example on the model is examined by the loss change at a reference point after one training step. Consider the model in the training step $i$ with parameters $\boldsymbol{\theta}_i$ and loss $\ell(\cdot; \boldsymbol{\theta})$ for some evaluation sample $\boldsymbol{z}'$. If we take a small gradient step on a training example $\boldsymbol{z}$, the first-order change in loss on $\boldsymbol{z}'$ can be approximated by the inner product between the reference gradient and the parameter update:

$$\ell(\boldsymbol{z}'; \boldsymbol{\theta}_{i+1}) - \ell(\boldsymbol{z}'; \boldsymbol{\theta}_i) \approx -\eta_i \left\langle \mathcal{U}\big(\nabla\ell(\boldsymbol{z}; \boldsymbol{\theta}_i), \boldsymbol{\theta}_i\big), \mathcal{U}\big(\nabla\ell(\boldsymbol{z}'; \boldsymbol{\theta}_i), \boldsymbol{\theta}_i\big) \right\rangle. \tag{7}$$

where $\mathcal{U}(\cdot, \boldsymbol{\theta}_i)$ is the optimizer update operator (e.g., SGD) and $\eta_i$ is the learning rate at step $i$. This inner product serves as a single-step influence score of the training sample $\boldsymbol{z}$ on the reference sample $\boldsymbol{z}'$, which essentially measures how the gradient of $\boldsymbol{z}$ aligns with the gradient of $\boldsymbol{z}'$. A higher positive value means that training on $\boldsymbol{z}$ would more greatly decrease the loss on $\boldsymbol{z}'$ (that is, $\boldsymbol{z}$ is beneficial for $\boldsymbol{z}'$), while a negative value means that $\boldsymbol{z}$ steps the model in a direction that increases the loss on $\boldsymbol{z}'$ (suggesting a conflict).

Because the model state evolves over the entire training process and the importance of specific data samples may vary across different stages, we examine the sample's influence trajectory over training. We divide training into $M$ snapshots $\{\boldsymbol{\theta}_1, \ldots, \boldsymbol{\theta}_M\}$. For a training example $\boldsymbol{z}$, define its update trajectory as the sequence of its update directions across these snapshots:

$$\mathrm{Traj}(\boldsymbol{z}; M) \triangleq \big\{\mathcal{U}\big(\nabla\ell(\boldsymbol{z}; \boldsymbol{\theta}_i), \boldsymbol{\theta}_i\big)\big\}_{i=1}^M. \tag{8}$$

This trajectory is essentially a series of gradient directions showing how $\boldsymbol{z}$ pushes the model at each stage of training. We can then quantify the cumulative influence of $\boldsymbol{z}$ on a reference example $\boldsymbol{z}'$ over the entire training process by summing the aligned influence at each stage:

$$\mathrm{Inf}^{\mathrm{Traj}}(\boldsymbol{z}, \boldsymbol{z}'; M) \triangleq \sum_{i=1}^M \bar{\eta}_i \langle \mathrm{Traj}(\boldsymbol{z}; i), \mathrm{Traj}(\boldsymbol{z}'; i) \rangle = \sum_{i=1}^M \bar{\eta}_i \langle \mathcal{U}\big(\nabla\ell(\boldsymbol{z}; \boldsymbol{\theta}_i), \boldsymbol{\theta}_i\big), \mathcal{U}\big(\nabla\ell(\boldsymbol{z}'; \boldsymbol{\theta}_i), \boldsymbol{\theta}_i\big) \rangle, \tag{9}$$

where $\bar{\eta}_i$ is the average learning rate in stage $i$.

## B.2 ADAMW TRAJECTORY INFLUENCE

VLMs fine-tuning often uses the AdamW optimizer (Loshchilov & Hutter, 2019), which includes decoupled weight decay. To accurately measure the influence under AdamW, we incorporate the weight decay into the gradient update vectors. We donate the update signal $\mathcal{U}_{\mathrm{AdamW}}\big(\nabla\ell(\boldsymbol{z}; \boldsymbol{\theta}_i), \boldsymbol{\theta}_i\big)$ for sample $\boldsymbol{z}$ in step $i$ as $\Psi(\boldsymbol{z}, \boldsymbol{\theta}_i)$:

$$\Psi(\boldsymbol{z}, \boldsymbol{\theta}_i) \triangleq \frac{\boldsymbol{m}^{t+1}}{\sqrt{\boldsymbol{v}^{t+1}} + \epsilon} + \lambda\boldsymbol{\theta}_i, \tag{10}$$

$$\boldsymbol{m}^{t+1} = \frac{\beta_1 \boldsymbol{m}^t + (1 - \beta_1)\nabla\ell(\boldsymbol{z}; \boldsymbol{\theta}_i)}{1 - \beta_1^t}, \tag{11}$$

$$\boldsymbol{v}^{t+1} = \frac{\beta_2 \boldsymbol{v}^t + (1 - \beta_2)(\nabla\ell(\boldsymbol{z}; \boldsymbol{\theta}_i))^2}{1 - \beta_2^t}, \tag{12}$$

where $\lambda$ denotes the weight decay coefficient, $\beta_1$ and $\beta_2$ are the hyperparameters for the first and second moments, respectively, and $\epsilon$ is a small constant. To reduce dimensionality, we first use Low-Rank Adaptation (LoRA) (Hu et al., 2022) to focus on a smaller set of parameters, with their update signal denoted $\hat{\Psi}(\boldsymbol{z}, \boldsymbol{\theta}_i)$. We then apply a fixed random projection (Johnson & Lindenstrauss, 1984) matrix $R \in \mathbb{R}^{d \times m}$ (where $m \ll d$) to project the high-dimensional update vectors into a $m$-dimensional space. The update signal in this scenario is given by

$$\tilde{\Psi}(\boldsymbol{z}; \boldsymbol{\theta}_i) = R^\top \hat{\Psi}(\boldsymbol{z}, \boldsymbol{\theta}_i). \tag{13}$$

Then the AdamW trajectory influence is:

$$\mathrm{Inf}_{\mathrm{AdamW}}^{\mathrm{Traj}}(\boldsymbol{z}, \boldsymbol{z}'; M) \triangleq \sum_{i=1}^M \bar{\eta}_i \cdot \cos\big(\tilde{\Psi}(\boldsymbol{z}; \boldsymbol{\theta}_i), \tilde{\Psi}(\boldsymbol{z}'; \boldsymbol{\theta}_i)\big) = \sum_{i=1}^M \bar{\eta}_i \cdot \frac{\langle \tilde{\Psi}(\boldsymbol{z}; \boldsymbol{\theta}_i), \tilde{\Psi}(\boldsymbol{z}'; \boldsymbol{\theta}_i) \rangle}{\|\tilde{\Psi}(\boldsymbol{z}; \boldsymbol{\theta}_i)\| \|\tilde{\Psi}(\boldsymbol{z}'; \boldsymbol{\theta}_i)\|}, \tag{14}$$

This influence measure $\mathrm{Inf}_{\mathrm{AdamW}}^{\mathrm{Traj}}(\boldsymbol{z}, \boldsymbol{z}')$ captures how similarly two samples $\boldsymbol{z}$ and $\boldsymbol{z}'$ drive the update of model parameters throughout the training.

## B.3 SELF-INFLUENCE

We define the self-influence (Bejan et al., 2023) of a training sample $z$ as the cumulative magnitude of its own gradient trajectory over training:

$$\mathrm{Inf}^{\mathrm{Self}}(\boldsymbol{z}; M) \triangleq \sum_{i=1}^M \bar{\eta}_i \langle \nabla\ell(\boldsymbol{z}, \boldsymbol{\theta}_i), \nabla\ell(\boldsymbol{z}, \boldsymbol{\theta}_i) \rangle, \tag{15}$$

which is essentially the sum of the squared gradient norms of $\boldsymbol{z}$ across all $M$ training stages (weighted by the learning rate at each stage). A higher self-influence $\mathrm{Inf}^{\mathrm{Self}}(\boldsymbol{z})$ means that $\boldsymbol{z}$ consistently produces large gradient updates – in other words, the model finds $\boldsymbol{z}$ difficult to learn. Self-influence thus serves as a proxy for how important or challenging a data point is.

## C  ALGORITHMIC

Algorithm 1 provides the algorithmic of the Capability-Attributed Data Curation (CADC) framework.

## D  TASK MAP

Table 8 lists the complete mapping of MMT-Bench (Ying et al., 2024) meta-tasks and subtasks to the three intrinsic capabilities discovered by CADC. The table provides detailed evidence for Section 4.3, illustrating how the categories defined by the benchmark diverge from the intrinsic capability structure.

Table 8 list the complete mapping of MMT-Bench meta-tasks and subtasks to the three intrinsic capabilities discovered by CADC. This mapping provides detailed evidence for §4.3, highlighting how the categories defined by the benchmark differ from the intrinsic capability structure. Furthermore, Figure 6 illustrates the distribution of subtasks across the three capabilities.

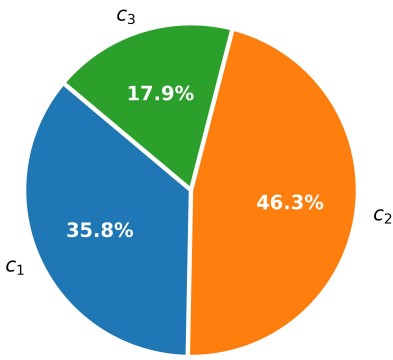

Figure 6: Proportion of the 162 subtasks assigned to each capability.

Table 8: Meta-tasks and subtasks grouped by intrinsic capability.

| Meta-Task | Subtask | # subtasks |
|---|---|---|
| $c_1$: Structural Grounding | | 58 |
| Keypoint Detection | Animal Keypoint Detection, Clothes Keypoint Detection, Furniture Keypoint Detection, Human Keypoint Detection, Vehicle Keypoint Detection | 5 |
| Localization | Camouflage Object Detection, Face Detection, Object Detection, Remote Sensing Object Detection, Rotated Object Detection, Salient Object Detection RGB, Salient Object Detection RGBD, Small Object Detection, Transparent Object Detection | 9 |
| Doc Understanding | Chart To Text, Table Structure Recognition, Visual Document Information Extraction | 3 |
| Counting | Counting By Category, Counting By Visual Prompting | 2 |
| Pixel Level Perception | Depth Estimation, Image Matting, Interactive Segmentation, Pixel Localization, Pixel Recognition, Polygon Localization | 6 |
| Anomaly Detection | Face Mask Anomaly Dectection | 1 |

Table 8 – continued from previous page

| Meta-Task | Subtask | # subtasks |
|---|---|---|
| Image Retrieval | Face Retrieval, Handwritten Retrieval, Image2image Retrieval, Person Reid, Sketch2image Retrieval, Text2image Retrieval, Vehicle Retrieval | 7 |
| Action Recognition | Gaze Estimation | 1 |
| Relation Reasoning | Human Interaction Understanding, Human Object Interaction Recognition, Scene Graph Recognition, Social Relation Recognition | 4 |
| Visual Captioning | Image Captioning Paragraph, Image Dense Captioning, Instance Captioning, Multiple Instance Captioning, Video Captioning, Writing Poetry From Image | 6 |
| Image Evaluation Judgement | Image Quality Assessment | 1 |
| Image-to-image Translation | Jigsaw Puzzle Solving | 1 |
| Meme Understanding | Meme Image Understanding, Meme Video Understanding | 2 |
| Temporal Understanding | Mevis, Next Img Prediction, Temporal Localization, Temporal Ordering | 4 |
| Embodied AI | Navigation | 1 |
| Cross Image Matching | One Shot Detection, Point Tracking, Single Object Tracking | 3 |
| Visual Grounding | Reason Seg, Referring Detection | 2 |
| $c_2$: Perceptual Recognition | | 75 |
| Visual Recognition | Abstract Visual Recognition, Age Gender Race Recognition, Animals Recognition, Animated Character Recognition, Astronomical Recognition, Building Recognition, Celebrity Recognition, Chemical Apparatusn Recognition, Color Recognition, Deepfake Detection, Disaster Recognition, Electronic Object Recognition, Fashion Recognition, Film and Television Recognition, Food Recognition, Gesture Recognition, Image Season Recognition, Landmark Recognition, Logo and Brand Recognition, Muscial Instrument Recognition, National Flag Recognition, Painting Recognition, Plant Recognition, Profession Recognition, Religious Recognition, Rock Recognition, Scene Recognition, Sculpture Recognition, Shape Recognition, Sports Recognition, Texture Material Recognition, Vehicle Recognition, Waste Recognition, Weapon Recognition, Weather Recognition | 35 |
| Action Recognition | Action Quality Assessment, General Action Recognition, Image Based Action Recognition, Sign Language Recognition | 4 |

Table 8 – continued from previous page

| Meta-Task | Subtask | # subtasks |
|---|---|---|
| Medical Understanding | Anatomy Identification, Disease Diagnose, Lesion Grading, Medical Modality Recognition, Other Biological Attributes | 5 |
| Emotion | Artwork Emotion Recognition, Body Emotion Recognition, Facial Expression Change Recognition, Facial Expression Recognition, Micro Expression Recognition, Scene Emotion Recognition | 6 |
| Hallucination | Attribute Hallucination, Exist Hallucination, Order Hallucination | 3 |
| Anomaly Detection | Behavior Anomaly Detection, Helmet Anomaly Detection, Industrial Produce Anomaly Detection, Traffic Anomaly Detection | 4 |
| Counting | Counting By Reasoning, Crowd Counting | 2 |
| Doc Understanding | Doc Vqa | 1 |
| OCR | Font Recognition, Scene Text Recognition | 2 |
| Visual Captioning | Image Captioning, Multiple Image Captioning | 2 |
| Image-to-image Translation | Image Colorization | 1 |
| Intelligence Quotient Test | Ravens Progressive Matrices | 1 |
| Multiple Image Analysis | Spot The Similarity | 1 |
| Temporal Understanding | Temporal Anticipation | 1 |
| Threed | Threed Cad Recognition, Threed Indoor Recognition | 2 |
| Autonomous Driving | Traffic Light Understanding, Traffic Participants Understanding, Traffic Sign Understanding | 3 |
| Visual Prompt Understanding | Visual Prompt Understanding | 1 |
| Visual Commonsense Reasoning | Whoops | 1 |
| $c_3$: Symbolic Reasoning | | 29 |
| Discipline Knowledge Reasoning | Art Design, Business, Health Medicine, Humanities Social Science, Science, Tech Engineering | 6 |
| Doc Understanding | Chart to Table, Chart VQA, Clock Reading | 3 |
| Visual Illusion | Color Assimilation, Color Constancy, Color Contrast, Geometrical Perspective, Geometrical Relativity | 5 |
| Visual Code | Eqn2latex, Screenshot2code, Sketch2code | 3 |
| GUI Navigation | Google Apps, GUI General, GUI Install, Web Shopping | 4 |
| OCR | Handwritten Mathematical Expression Recognition, Handwritten Text Recognition | 2 |
| Image Evaluation Judgement | LVLM Response Judgement | 1 |

Table 8 – continued from previous page

| Meta-Task | Subtask | # subtasks |
|---|---|---|
| Autonomous Driving | Multiple View Image Understanding, Temporal Sequence Understanding | 2 |
| Hallucination | Relation Hallucination | 1 |
| Visual Prompt Understanding | Som(Set-of-marks) Recognition | 1 |
| Multiple Image Analysis | Spot The Diff | 1 |

# E    TRAINING DATASET SAMPLING

To support capability-aware allocation, we analyze the sampling distribution of the Mix665K instruction-tuning dataset under the CADC framework. Each sample is attributed to one or more of the three intrinsic capabilities — structural grounding ($c_1$), perceptual recognition ($c_2$), and symbolic reasoning ($c_3$) — based on self-influence analysis. This attribution enables us to quantify how the training data are distributed across capabilities and to guide the construction of balanced and sequenced curricula.

Figure 7 summarizes the sampling statistics. The left panel shows the overall proportion of samples assigned to each capability cluster, while the right panel breaks down the composition into capability-exclusive samples and samples shared across multiple capabilities.

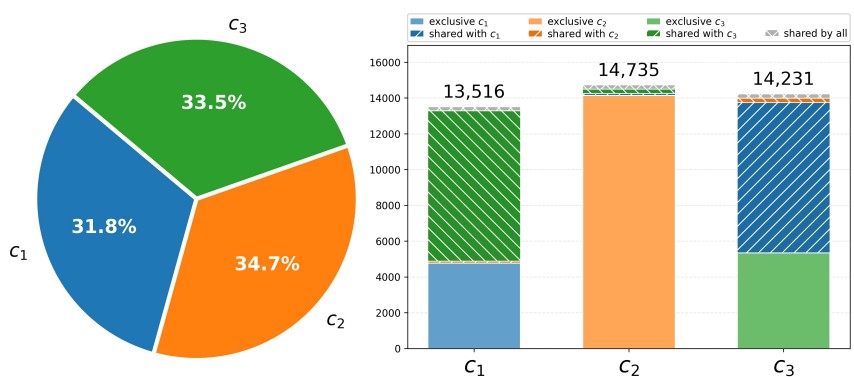

Figure 7: Sampling statistics of the Mix665K instruction-tuning dataset across the three capabilities discovered by CADC. Left: proportions of samples assigned to each capability. Right: sample composition, showing capability-exclusive data and samples shared across capabilities.

Table 9 reports the corresponding counts. For each capability, the table lists exclusive samples, overlaps with other clusters, multi-capability samples, and totals. These values serve as the basis for the allocation of sample budgets between capabilities (§3.3.1) and ensure the reproducibility of our experimental setup.

# F    EXPERIMENTAL SETUP DETAILS

**Training datasets.** We follow Wu et al. (2024) and adopt the LLaVA-1.5 Mix665K (marked as Mix665K) instruction-tuning dataset (Liu et al., 2024a). This dataset contains approximately 665K examples, combining GPT-generated samples (Liu et al., 2023) with existing resources such as TextCaps (Sidorov et al., 2020) and VG (Krishna et al., 2017).

**Evaluation datasets.** We evaluate our method on two benchmark suites. (1) The first comprises LLaVA-Wild Bench (Liu et al., 2023), VQAv2 (Goyal et al., 2017), POPE (Li et al., 2023b), MM-Bench (Liu et al., 2024b), ScienceQA (Lu et al., 2022) and SEED-Bench (Li et al., 2023a), following ICONS (Wu et al., 2024) and TIVE (Liu et al., 2024c), and evaluated with the LMMs-Eval (Zhang et al., 2025) framework. (2) The second includes RealWorldQA (xAI, 2024), HallusionBench (Guan

---

**Algorithm 1:** Capability-Attributed Data Curation (Expanded Version)

---

**Input:** Training dataset $\mathcal{D}_{\text{train}}$, target dataset $\mathcal{D}_{\text{target}}$ with subtask set $\mathcal{T}$, model snapshots $\{\boldsymbol{\theta}_1, \ldots, \boldsymbol{\theta}_M\}$, threshold $\tau$, tolerance $\delta$, and budget $N$

**Output:** Curated curriculum $\widetilde{\mathcal{D}}_{\text{train}}$

1 **Phase 1: Capability Discovery**

2 **foreach** $\boldsymbol{z}' \in \mathcal{D}_{target}$ **do**

3     Record AdamW update signal:

$$\tilde{\Psi}(\boldsymbol{z}'; \boldsymbol{\theta}_i) = R^{\top} \hat{\Psi}(\boldsymbol{z}'; \boldsymbol{\theta}_i), \quad i = 1, \ldots, M,$$

    where $\hat{\Psi}$ is the LoRA-projected AdamW update and $R$ is a fixed random projection.

4 For each subtask $t \in \mathcal{T}$, compute its trajectory vector:

$$\overline{\text{Traj}}(t) = \sum_{i=1}^{M} \bar{\eta}_i \, \mathbb{E}_{\boldsymbol{z}' \sim t} \big[ \tilde{\Psi}(\boldsymbol{z}'; \boldsymbol{\theta}_i) \big].$$

5 Build similarity graph $\mathcal{G} = (V, E)$ with $V = \mathcal{T}$ and

$$(t_m, t_n) \in E \quad \Leftrightarrow \quad \cos(\overline{\text{Traj}}(t_m), \overline{\text{Traj}}(t_n)) > \tau.$$

6 Apply Leiden community detection:

$$\mathcal{C} = \{c_1, \ldots, c_K\}, \quad \{\mathcal{D}_{\text{target}}^{(k)}\}_{k=1}^{K} \leftarrow \text{CommunityDetection}(\mathcal{G}).$$

7 **Phase 2: Capability Attribution**

8 **foreach** $\boldsymbol{z} \in \mathcal{D}_{train}$ **do**

9     For each capability $c_k$ with target subset $\mathcal{D}_{\text{target}}^{(k)}$, compute:

$$\text{Inf}^{\text{Traj}}(\boldsymbol{z}, c_k) = \mathbb{E}_{\boldsymbol{z}' \in \mathcal{D}_{\text{target}}^{(k)}} \left[ \sum_{i=1}^{M} \bar{\eta}_i \frac{\langle \tilde{\Psi}(\boldsymbol{z}; \boldsymbol{\theta}_i), \tilde{\Psi}(\boldsymbol{z}'; \boldsymbol{\theta}_i) \rangle}{\|\tilde{\Psi}(\boldsymbol{z}; \boldsymbol{\theta}_i)\| \cdot \|\tilde{\Psi}(\boldsymbol{z}'; \boldsymbol{\theta}_i)\|} \right].$$

10     Assign $\boldsymbol{z}$ into $\mathcal{D}_{\text{train}}^{(k)}$ if

$$\max_{j} \text{Inf}^{\text{Traj}}(\boldsymbol{z}, c_j) - \text{Inf}^{\text{Traj}}(\boldsymbol{z}, c_k) \leq \delta.$$

11 **Phase 3: Curriculum Curation**

12 Define self-influence for each capability:

$$\text{Inf}^{\text{Self}}(c_k) = \mathbb{E}_{\boldsymbol{z} \in \mathcal{D}_{\text{train}}^{(k)}} \left[ \sum_{i=1}^{M} \bar{\eta}_i \langle \nabla \ell(\boldsymbol{z}; \boldsymbol{\theta}_i), \nabla \ell(\boldsymbol{z}; \boldsymbol{\theta}_i) \rangle \right].$$

13 Allocate budget:

$$N_k = \frac{\text{Inf}^{\text{Self}}(c_k)}{\sum_{j=1}^{K} \text{Inf}^{\text{Self}}(c_j)} \cdot N.$$

14 **foreach** $c_k \in \mathcal{C}$ **do**

15     Select $\text{Top}_{N_k}$ samples in $\mathcal{D}_{\text{train}}^{(k)}$ by $\text{Inf}^{\text{Traj}}(\boldsymbol{z}, c_k)$.

16 Infer curriculum order $(c_{i_1} \prec c_{i_2} \prec \cdots \prec c_{i_K})$ from temporal self-influence dynamics.

17 Schedule staged training: in phase $j$, focus on $\mathcal{D}_{\text{train}}^{(i_j)}$ with replay of earlier phases.

18 **return** $\widetilde{\mathcal{D}}_{train}$

---

Table 9: Distribution of Mix665K training samples across intrinsic capabilities. Columns report exclusive counts, overlaps with other capabilities, multi-capability samples, and totals.

| Capability | Exclusive | Shared with | | | | Total |
|---|---|---|---|---|---|---|
| | | $c_1$ | $c_2$ | $c_3$ | Multi | |
| $c_1$ | 4,776 | – | 108 | 8,400 | 232 | 13,516 |
| $c_2$ | 14,151 | 108 | – | 244 | 232 | 14,735 |
| $c_3$ | 5,355 | 8,400 | 244 | – | 232 | 14,231 |

et al., 2024), ScienceQA (Lu et al., 2022), TextVQA (Singh et al., 2019), DocVQA (Mathew et al., 2021), and MMT-Bench (Ying et al., 2024), assessed using the VLMEvalKit (Duan et al., 2024) framework. For SEED, we evaluate only its image subset, while for SQA we report results on both the full benchmark and its image subset. For MMBench, we report results for both the English (EN) and Chinese (CN) variants. Finally, MMT has two variants: MMT-SI, where all images from a single entry are merged into one, and MMT-MI, where images remain unmerged. Collectively, these benchmarks span a wide range of formats and objectives, covering tasks such as recognition, reasoning, and hallucination.

**Target datasets.** Unlike previous approaches that draw target data from multiple evaluation benchmarks (Xia et al., 2024; Wu et al., 2024), CADC designates only the validation split of MMT-Bench (Ying et al., 2024), a comprehensive multimodal benchmark comprising 162 subtasks, as the target dataset $\mathcal{D}_{\text{target}}$ (§3.2).

**Models for data selection and training.** We adopt SmolVLM-256M (Marafioti et al., 2025), a compact open-source vision–language model, as the data selection model instead of the commonly used LLaVA-v1.5-7B (Liu et al., 2024a). Large models primarily acquire their core knowledge during pretraining, while instruction tuning serves to unlock and align these capabilities (Zhou et al., 2023; Xia et al., 2024). However, LLaVA is pretrained on only 558K samples, less than its 665K instruction-tuning samples, indicating that its pretraining is insufficient, making it unreliable as an analytical tool for the instruction-tuning phase. For training, we also use SmolVLM-256M, while also employing LLaVA-v1.5-7B to ensure comparability with previous studies.

**Default setting.** We record $M = 4$ snapshots of the data selection model. In experiments with SmolVLM-256M, the similarity threshold for edge generation is set to $\tau = 0.2$ (§3.1), and the tolerance for data attribution is set to $\delta = 0.01$ (§3.2). For the training models, we strictly follow their original configurations (e.g., learning rate, optimizer) to ensure controlled comparisons.

## G BASELINES DETAILS

To ensure a thorough and fair evaluation, our CADC is benchmarked against a wide range of existing data selection methods. These baselines can be grouped into heuristic methods (**Random**, **Length**, **Perplexity** (Marion et al., 2023)), embedding-based approaches (**CLIP-Score** (Radford et al., 2021), **D2-Pruning** (Maharana et al., 2024), **COINCIDE** (Lee et al., 2024)), gradient-driven scores (**EL2N** (Paul et al., 2021), **GraNd** (Paul et al., 2021)), self-supervised strategies (**Self-Sup** (Sorscher et al., 2022), **Self-Filter** (Chen et al., 2024)), and influence-driven methods (**LESS** (Xia et al., 2024), **TIVE** (Liu et al., 2024c), **ICONS** (Wu et al., 2024)). In the following, we summarize each method with its main principle and context of use.

- **Random** selects samples uniformly at random, serving as a simple but effective baseline.

- **Length** prioritizes samples with longer instructions, under the assumption that they contain richer information.

- **Perplexity** (Marion et al., 2023) scores samples according to next-token prediction uncertainty, where higher perplexity suggests greater learning difficulty and potential utility.

- **CLIP-Score** (Radford et al., 2021) uses CLIP embeddings to measure image-text alignment, selecting samples with higher alignment scores.

Table 10: Performance of SmolVLM under different data selection methods. *Rel. Avg.* is relative to the full-data baseline (100%). $c_1$, $c_2$, and $c_3$ denote the model's performance on structural grounding, perceptual recognition, and symbolic reasoning, respectively.

| Method | Data % | Sel. Model | Size | RWQA↑ | Hallusion↑ | SQA↑ | TextVQA↑ | Doc↑ | MMT-SI↑ | MMT-MI↑ | Rel. Avg.↑ | $c_1$↑ | $c_2$↑ | $c_3$↑ |
|---|---|---|---|---|---|---|---|---|---|---|---|---|---|---|
| LESS | 5% | SmolVLM | 256M | 30.6 | 21.7 | 35.5 | 32.8 | 6.2 | 23.4 | 23.9 | 91.5% | 21.1 | 27.2 | 18.3 |
| ICONS | 5% | SmolVLM | 256M | 32.3 | 18.1 | 35.0 | 31.8 | 8.0 | 23.2 | 23.6 | 89.2% | 20.0 | 26.6 | 21.2 |
| TIVE | 5% | SmolVLM | 256M | 37.0 | 18.1 | 36.1 | 30.8 | 15.0 | 24.1 | 24.6 | 96.5% | 20.4 | 26.7 | 25.9 |
| CADC (our) | 5% | SmolVLM | 256M | 42.7 | 24.1 | 35.5 | 33.5 | 16.5 | 25.5 | 25.7 | 107.1% | 22.1 | 27.2 | 26.2 |

- **D2-Pruning** (Maharana et al., 2024) employs graph-based pruning to maximize diversity while maintaining the representativeness of the data.

- **COINCIDE** (Lee et al., 2024) uses a smaller reference model to cluster data by concept–skill compositions, sampling for both diversity and transferability.

- **EL2N** (Paul et al., 2021) estimates sample importance by computing the L2-norm of the error vector across tokens, widely used in image classification and adapted here for vision–language instruction data.

- **GraNd** (Paul et al., 2021) scores samples using the L2-norm of gradients induced by each training example, reflecting their potential contribution to parameter updates.

- **Self-Sup** (Sorscher et al., 2022) selects prototypical samples by unsupervised clustering, aiming to represent the overall distribution.

- **Self-Filter** (Chen et al., 2024) trains a scoring network jointly with a reference LVLM to filter instruction data based on learned quality signals.

- **LESS** (Xia et al., 2024) adapts influence estimation to Adam optimization and variable length instructions, enabling efficient Low-rank gradient similarity search for targeted instruction tuning.

- **TIVE** (Liu et al., 2024c) scores instances based on both influence and task difficulty, pruning redundant data while preserving high-value samples for visual instruction tuning.

- **ICONS** (Wu et al., 2024) selects data through cross-task influence consensus, identifying samples consistently valuable across tasks through majority voting over influence matrices.

Together, these baselines cover the major paradigms in data selection, from simple heuristics and embedding-based filtering to advanced gradient- and influence-driven strategies. This breadth ensures that CADC is evaluated not only against lightweight heuristics but also against SOTA influence-based methods specifically developed for instruction tuning.

## H  MORE EXPERIMENT RESULTS

### H.1  BASELINE COMPARISON ON SMOLVLM AND CAPABILITY-LEVEL EVALUATION

To complement the main results, we provide a detailed baseline comparison on the SmolVLM-256M model in a unified training environment. Table 10 reports the performance of CADC against recent data selection methods, including LESS, ICONS, and TIVE. This perspective highlights how CADC achieves consistent improvements over previous approaches even with the compact SmolVLM model, underscoring the robustness and generality of our framework.

Beyond general benchmark scores, we further analyze performance at the level of intrinsic capabilities. Specifically, the evaluation results of the subtasks in MMT-Bench are grouped according to the mapping in Table 8, and the mean value within each group is computed. These averages serve as performance estimates for the three discovered capabilities: $c_1$ (structural grounding), $c_2$ (perceptual recognition), and $c_3$ (symbolic reasoning). This decomposition enables us to quantify how each selection method influences distinct capabilities, offering a more fine-grained understanding of their effects.

## H.2 Component Ablation

To evaluate the importance of each component in CADC, we perform ablation experiments on SmolVLM-256M, with results summarized in Table 11. We examine the following variants:

Table 11: Ablation study on the CADC.

| Method | RWQA↑ | Hallusion↑ | SQA↑ | TextVQA↑ | Doc↑ | MMT-SI↑ | MMT-MI↑ | Rel. Avg.↑ |
|---|---|---|---|---|---|---|---|---|
| **CADC** | 42.7 | **24.1** | 35.5 | **33.5** | 16.5 | **25.5** | **25.7** | **107.1%** |
| w/o capability | 45.8 | 20.8 | 35.4 | 32.6 | **16.8** | 24.6 | 25.0 | 93.2% |
| w/o budget allocation | 38.2 | 18.9 | 34.5 | 31.9 | 16.7 | 23.1 | 22.6 | 104.0% |
| w/o pool sampling | **47.6** | 20.5 | **35.8** | 32.4 | 7.9 | 23.8 | 24.0 | 96.6% |
| w/o sequence | 33.7 | 15.5 | 34.0 | 31.1 | 16.7 | 24.2 | 23.9 | 97.4% |

- **w/o capability.** Capability discovery is disabled; instead, the data is grouped by manually defined task labels rather than by intrinsic capabilities.
- **w/o budget allocation.** Sampling budgets are distributed equally across all groups, ignoring demand signals provided by self-influence.
- **w/o pool sampling.** Training data are drawn uniformly at random from each capability pool, eliminating prioritization of trajectory influence.
- **w/o sequence.** All selected data are provided to the model at once in random order, omitting the curriculum sequencing guided by self-influence dynamics.

This study isolates the contribution of each design choice. As shown in the table, the removal of any single component consistently reduces overall performance, with huge drops observed when capability discovery is removed, highlighting its central role in CADC's effectiveness.

## I Ethics statement

This work focuses on a methodology to improve the efficiency of supervised fine-tuning in large vision–language models (VLMs). No human subjects, personally identifiable information, or sensitive user data were involved in any stage of the research. All experiments were conducted on publicly available benchmarks (e.g., LLaVA-Wild, ScienceQA, and MMT-Bench) that have already undergone community vetting. The proposed framework, Capability-Attributed Data Curation (CADC), is designed for general research purposes and does not inherently produce harmful content. However, like all model optimization methods, it could be misapplied to domains with potential ethical risks (e.g., misinformation or surveillance). We therefore emphasize that CADC should only be used in accordance with responsible AI practices and the ICLR Code of Ethics. All authors affirm that there are no conflicts of interest, sponsorship biases, or ethical violations associated with this study.

## J Reproducibility statement

We have made efforts to ensure the reproducibility of our work. The methodology is described in detail in Section 3, with precise definitions of intrinsic capability discovery (§3.1), attribution (§3.2), and curation (§3.3). Hyperparameters such as snapshot count $M$, similarity threshold $\tau$, and tolerance $\delta$ are reported in experimental settings. All benchmarks, datasets, and baselines are publicly available. Additional implementation details, derivations, and sampling strategies are provided in the appendices (Appendices B–E). The results are presented in multiple models (SmolVLM variants and LLaVA-v1.5), datasets (Mix665K, Vision-Flan), and ablations (Table 3), to demonstrate robustness. The source code and the curated data splits will be released upon acceptance to facilitate independent verification.

## K Declaration of Large Language Model Use

In the preparation of this manuscript, ChatGPT-5 was used solely to aid and polish the writing. Specifically, the LLM was used to improve clarity, conciseness, and adherence to academic conventions in the English text. The LLM did not conduct any part of the research design, data analysis,

results generation, or interpretation of the findings. All scientific contributions and intellectual content are the responsibility of the authors.

