# OpenReview forum: "Uncovering Intrinsic Capabilities: A Paradigm for Data Curation in Vision-Language Models"
_ICLR.cc/2026/Conference — Submitted to ICLR 2026_

### Official Review · Reviewer_xEJf · 2025-10-27

**Soundness:** 2
**Presentation:** 2
**Contribution:** 2
**Rating:** 4
**Confidence:** 4

**Summary:**

This paper addresses the challenge of data curation for instruction tuning in large vision-language models (VLMs). The authors argue that conventional heuristic-based data reduction strategies treat models as "black boxes" and often lead to performance degradation. To solve this, the paper introduces the CADC framework. CADC shifts the paradigm from task-specific heuristics to an analysis of the "intrinsic capabilities" that the model develops during the learning process.

**Strengths:**

1. CADC actually introduces a novel method for guiding model training and evolution by analyzing gradient-based learning trajectories on multi-task benchmarks to understand and leverage a model's intrinsic capabilities.

2. CADC can achieve comparable or even superior performance to training on a full dataset while using as little as 5% of the original instruction tuning data.

**Weaknesses:**

1. Questionable Generalizability to Large-Scale Models: The framework's core experiments are conducted on a very small model (SmolVLM-256M). While showing benefits on this scale, the claim of achieving superior performance with only 5% of data may not hold for current state-of-the-art, large-scale Vision-Language Models (e.g., LLaVA-OV, Qwen-VL 2.5), whose learning dynamics and data requirements are substantially different.

2. Although the final training set is smaller, the process to create it is computationally expensive. The framework requires calculating gradient-based trajectories and influence scores for the entire dataset, a process far more costly than simpler data selection heuristics. This high upfront cost of curation might offset the efficiency gains made during the final fine-tuning stage.

3. The framework distills 162 distinct subtasks from MMT-Bench into just three broad intrinsic capabilities. This high level of abstraction risks oversimplification, potentially masking crucial, finer-grained skills. For instance, grouping dozens of diverse tasks under "Perceptual Recognition" may prevent the specialized curation needed for distinct sub-skills within that category, leading to a less nuanced training curriculum.

**Questions:**

1. On Model Generalizability: How does the framework's effectiveness, particularly the 5% data efficiency, generalize to current state-of-the-art models significantly larger than 7B parameters (e.g., the Qwen-VL 2 / 2.5 series)? Does the performance advantage of CADC diminish as the base model becomes more capable?

2. Could you provide a detailed analysis of the computational overhead for the data selection stage? More importantly, what is the underlying reason for the performance degradation observed when the data subset was increased from 15% to 20%?

3. What is the justification for selecting exactly three intrinsic capabilities (K=3)? Could you provide an ablation study on K to show how performance varies with different numbers of clusters? Furthermore, are these discovered capabilities a stable property of the model, or an artifact of using MMT-Bench as the target dataset?

---

> ### Author Response · Authors · 2025-11-26
>
> We thank Reviewer xEJf for the helpful comments on generalization, efficiency, and capability design. Based on feedback from all reviewers, we have added new experiments and clarifications to further strengthen the paper.
>
> **Summary of Major Updates**
>
> - **Capability robustness.** We added analyses to test the stability of capability discovery, including sweeping the task-graph connectivity threshold and replacing MMT-Bench with ICONS as the target suite. Across these variants, CADC consistently discovers three capabilities and maintains strong performance.
> - **Computational cost.** We introduced a dedicated complexity analysis that decomposes the offline cost of CADC and shows that the overall selection overhead is practically acceptable. We also study variants with fewer snapshots, which further reduce cost while preserving most of the performance gains.
> - **Scaling to larger backbones.** We extended our experiments to Qwen2-VL models up to 72B parameters, all trained on the same 5% subset selected once by CADC on SmolVLM-256M. The CADC-curated subset matches or exceeds the full-data baseline while requiring substantially lower training time for the largest model.
>
> We address the reviewer's specific points below.
>
> ## Weakness 2; Question 2
>
> We agree that the computational cost of the data selection stage is important for assessing CADC’s practicality.
>
> 1. **Computational cost.** In the revised version, we add Sec. 4.2.4 with a dedicated complexity analysis and report the wall-clock cost of CADC’s data processing pipeline in Table 6.
>     - This curation stage is run **once** to produce a curated subset that can then be reused across multiple training runs and backbones, so the overhead is a **one-time offline cost** that can be amortized rather than paid again for every model.
>     - Table 7 further shows that using **fewer snapshots** substantially reduces the trajectory overhead while CADC still preserves most of its gains and continues to outperform the 100%-data baseline, demonstrating that CADC can operate under tighter compute budgets by trading a modest performance drop for a significant reduction in curation cost.
>     - Combined with Fig. 4, which shows that for Qwen2-VL models the total cost of “selection with a small model + training on 5% CADC data” is lower than training the large model on 100% of the data while achieving better or comparable performance, this indicates that the upfront curation cost does not negate—and in large-model regimes can actually enhance—the net efficiency gains.
>     - Compared to heuristic selectors, CADC avoids the **implicit cost of manually designing and tuning capability definitions and scoring rules**, since both the capability structure and scores are derived automatically from the model’s learning dynamics rather than from hand-crafted heuristics.
>
>     *Table 7: Performance of CADC under different snapshot counts .*
>
>     |  | **Rel. Avg.↑** |
>     | --- | --- |
>     | Random | 96.4% |
>     | M = 1 | 106.4% |
>     | M = 4 (default) | **107.1%** |
> 2. **The 15% vs. 20% behavior.** We note that both 15% and 20% still outperform training on 100% of the data; the “degradation” is only a slightly smaller **relative** margin at 20%. This is a standard diminishing-returns effect under a fixed full-data baseline: at 15%, CADC can focus almost entirely on the highest-value examples, while moving to 20% inevitably introduces lower-value or even distracting samples, which can mildly interfere with learning and shrink the margin [1,2,3]. Intuitively, once a carefully selected 5–15% subset already beats the 100% data, increasing the fraction must eventually bring in examples that look more like the average of the full dataset, so performance drifts back toward the 100% baseline rather than growing monotonically above it. This behavior indicates that CADC has already concentrated most of the valuable training signal into the smaller subsets, and the remaining data are largely lower-quality or redundant—exactly what one would expect from an effective data curation method.
>
> [1] Zhou Q, et al. “Scale Efficient Training for Large Datasets.” CVPR, 2025.
>
> [2] Askari-Hemmat R, et al. “Improving the Scaling Laws of Synthetic Data with Deliberate Practice.” *arXiv preprint arXiv:2502.15588* (2025).
>
> [3] Ankner Z, et al. “Perplexed by perplexity: Perplexity-based data pruning with small reference models.” ICLR, 2025.

---

> > ### Author Response · Authors · 2025-11-26
> >
> > ## Weakness 1; Question 1
> >
> > We agree that it is important to assess whether CADC remains effective beyond a small model like SmolVLM-256M. In fact, our experiments already span a wide range of model scales. CADC selects 5% of Mix665K using SmolVLM-256M as the data selection model.
> >
> > - In the original submission, we train **SmolVLM-256M**, **SmolVLM-2.2B** (Table 2), and **LLaVA-v1.5-7B** (Table 1) on this same 5% subset.
> > - In the revised version, **Fig. 4** further evaluates **Qwen2-VL** models at **2B, 7B, and 72B** parameters on exactly this 5% CADC subset.
> >
> > Across all these backbones—from 256M to 2B, 7B, and 72B—the 5% CADC subset consistently **matches or exceeds** the performance of training on 100% of the data. We therefore do not observe a diminishing advantage as the base model becomes more capable; if anything, stronger models are able to exploit the curated subset at least as well as smaller ones. These results indicate that CADC is capturing stable structure in the learning dynamics that transfers across model scales, rather than artifacts of a particular small backbone, and support our claim that CADC improves data efficiency even for modern, large VL models.
> >
> > ## Weakness 3; Question 3
> >
> > We understand the concern that distilling 162 subtasks into three capabilities might risk oversimplification. In CADC, however, both how $K$ is obtained and how capabilities are used are driven by the model’s learning dynamics rather than manual design:
> >
> > 1. **How $K=3$ is obtained.** We do **not** set $K$ as a hyperparameter. CADC first builds a task graph from similarities between gradient trajectories and then applies community detection. As shown in Table 4 of the revised paper, for a **wide range** of graph connectivity thresholds $\tau \le 0.25$, this procedure consistently recovers **three** communities and achieves the best performance in that regime. When $\tau$ is pushed higher so that $K>3$, performance does not improve and moves closer to the “w/o Cluster” and human task-grouping baselines (capability ablation of Table 3). This suggests that three capabilities already capture the main structure relevant for training, and that further splitting mainly introduces fragmentation and noise rather than uncovering useful extra skills.
> >
> >     *Table 4: Effect of the task-graph threshold τ on cluster count and relative performance.*
> >
> >     | **Threshold** | $\tau=0$ | $\tau=0.05$ | $\tau=0.1$ | $\tau=0.15$ | $\tau=0.2$ | $\tau=0.25$ | $\tau=0.3$ | $\tau=0.35$ | $\tau=0.45$ | w/o Cluster |
> >     | --- | --- | --- | --- | --- | --- | --- | --- | --- | --- | --- |
> >     | **# Cluster** | 3 | 3 | 3 | 3 | 3 | 3 | 4 | 7 | 12 | 1 |
> >     | **Rel. Avg.↑** | 99.7% | 99.3% | 97.1% | 99.8% | **100.0%** | 99.7% | 96.7% | 97.6% | 97.1% | 95.8% |
> > 2. **Abstraction level.** Our goal is not to propose a new manual task taxonomy, but to identify a **small number of latent skill axes** along which the model’s learning dynamics meaningfully differ, so that data curation can ensure coverage of each axis instead of treating all tasks as a flat, undifferentiated list. The three discovered capabilities play exactly this role: they summarize how tasks co-evolve during training and support a tractable capability-wise curriculum. Empirically, the radar plots in Fig. 1 show that CADC improves or maintains performance on **all** capabilities without creating new blind spots, indicating that this abstraction is sufficient to guide more balanced and efficient data usage.
> > 3. **Stability beyond MMT-Bench.** The capability structure is not an artifact of MMT-Bench. In the revised Table 5, we **replace MMT-Bench with the ICONS task set** when constructing the task graph and again stably obtain **three** capability groups, while CADC reaches 102.8% of the full-data performance on this ICONS target. This consistency across different target suites suggests that the three capabilities reflect a stable property of the model’s behavior, rather than being tied to a particular benchmark definition.
> >
> >     *Table 5: Cluster count and relative performance when discovering capabilities on different target datasets.*
> >
> >     | **Method** | **Target Datasets** | **# Cluster** | **Rel. Avg.↑** |
> >     | --- | --- | --- | --- |
> >     | **Random** | - | - | 96.4% |
> >     | **CADC** | MMT-Bench | 3 | **107.1%** |
> >     | **CADC** | ICONS | 3 | 102.8% |
> >
> > We hope that the additional experiments and clarifications address your concerns, and we would be grateful if you could take them into account when finalizing your overall assessment and recommendation.

---

> ### Comment · Reviewer_xEJf · 2025-11-26
>
> Thanks for clarification. This has addressed most of my concerns. So i will rise my score to 6.

---

> > ### Author Response · Authors · 2025-11-26
> >
> > We sincerely appreciate your constructive feedback and are grateful for your decision to raise your score. We will incorporate the discussed revisions into the final manuscript. Thank you again for helping us improve our paper.

---

### Official Review · Reviewer_KJpr · 2025-10-27

**Soundness:** 2
**Presentation:** 3
**Contribution:** 2
**Rating:** 4
**Confidence:** 4

**Summary:**

This paper proposes a data selection method for MLLMs that uses gradient-based learning trajectories of MMT-Bench as an influence metric. The method clusters validation data into three high-level categories (grounding, perception, reasoning) based on the cosine similarity of these trajectories. It then uses this clustering to adapt the data budget allocation and the training data sequence, framing this as a score-based curriculum learning approach.

**Strengths:**

1. The introduction of a score-based curriculum learning framework for the MLLM data selection task is a promising direction to the field.

2. The proposed data selection pipeline is comprehensive, addressing the full process from data scoring and attribution to data budget allocation and the final learning scheme design.

**Weaknesses:**

1. The motivation for the paper feels insufficiently justified, and the novelty of the proposed method appears limited. For example, the manuscript fails to clearly differentiate its core concept of "capability" from existing concepts, such as the "concepts" or "skills" discussed in prior work (e.g., COINCIDE). While the specific score formulation may differ, the fundamental method of performing data attribution via clustering is highly similar. The proposed approach could be interpreted as a simplified/coarse instance of this existing framework, with a K=3 clustering. The claim that this K=3 clustering makes the method "white-box" or "intrinsic" to the model dynamics is not well-supported. Furthermore, the three identified categories (perception, reasoning, etc.) are already widely acknowledged in the literature (e.g., in the MME benchmark's level-1 categories) and do not offer new insight.

2. There appears to be a contradiction in the paper's positioning. The authors claim their method is distinct from previous heuristic and task-driven approaches, yet they utilize MMT-Bench as a validation set to discover capabilities. This discovery process itself appears to be heuristic, which undermines the initial claim.

3. The authors' end-to-end pipeline includes a Sequence training stage. It is unclear if the baseline methods used for comparison also benefit from this sequencing stage. If they do not, the reported performance gains may stem from the curriculum learning scheme rather than the novel data selection metric itself. This must be clarified to validate the paper's core claims.

4. The paper is missing a computational analysis of the entire pipeline. Calculating gradient-based learning trajectories is known to be computationally expensive. This is a critical omission for a data selection method, especially one based on influence functions. The authors should provide a thorough analysis of the computational overhead (e.g., similar to Figure 7 in COINCIDE) to assess the method's practical feasibility.

5. The presentation of benchmark results is inconsistent across the manuscript's tables and figures. This inconsistency makes it difficult for the reviewer to follow the experiments and accurately assess the method's efficacy.

**Questions:**

Heuristic-based finer-grained capabilities works with the advantage of being more scalable and are demonstrably effective and generalizable. For example, Harvesting Rich, Scalable and Transferable Multi-Modal Data for Instruction Fine-Tuning (2025). Can we reasonably assume that existing heuristic approaches remain a more practical solution regarding computational cost? The authors should provide a detailed computational analysis to correct this assumption if it is wrong.

---

> ### Author Response · Authors · 2025-11-26
>
> We thank Reviewer KJpr for their insightful feedback and for raising important conceptual questions about our approach. Based on the valuable comments from all reviewers, we have made three major updates and prepared detailed clarifications to strengthen our work.
>
> **Summary of Major Updates**
>
> - **Capability robustness.** We added analyses to test the stability of capability discovery, including sweeping the task-graph connectivity threshold and replacing MMT-Bench with ICONS as the target suite. Across these variants, CADC consistently discovers three capabilities and maintains strong performance.
> - **Computational cost.** We introduced a dedicated complexity analysis that decomposes the offline cost of CADC and shows that the overall selection overhead is practically acceptable. We also study variants with fewer snapshots, which further reduce cost while preserving most of the performance gains.
> - **Scaling to larger backbones.** We extended our experiments to Qwen2-VL models up to 72B parameters, all trained on the same 5% subset selected once by CADC on SmolVLM-256M. The CADC-curated subset matches or exceeds the full-data baseline while requiring substantially lower training time for the largest model.
>
> We address the reviewer's specific points below.
>
> ## Weakness 1
>
> We address the reviewer’s points in three parts.
>
> 1. **Distinction from COINCIDE’s concepts/skills.**
> In COINCIDE, “concepts” and “skills” are used as an informal narrative for the selection score: “skills” are neither formally defined nor studied as the main object. In CADC, *intrinsic capabilities* are central and formally defined in Sec. 2 and App. A as “a latent skill such that performance on any task can be factorized into contributions from one or more of these capabilities,” and we explicitly analyze performance and behavior at the capability level (Fig. 1 right, Sec. 4.3). Methodologically, COINCIDE clusters **training examples** and uses the resulting clusters as selection units; CADC clusters **tasks** once to discover a small set of capabilities, and then performs attribution and selection *with respect to those capabilities* rather than over example clusters. Thus both the underlying construct and the role of clustering differ substantially, and CADC is not simply “data attribution via clustering” in the COINCIDE sense. And CADC is not just a coarse $K=3$ instance of COINCIDE, which heuristically sets $K = 10{,}000$, yet achieves weaker results in our shared settings (Table 1(b),Fig. 3(a)).
> 2. **On “$K = 3$” and being white-box.**
>
>     Our notion of “intrinsic” refers to decomposing a black-box model into latent skills from its own learning dynamics, not to manually fixing $K=3$. In CADC, $K$ is never set by hand: the three capabilities emerge from the task-graph structure, and as shown in the revised Table 4, this $K=3$ solution is stably recovered across a wide range of thresholds with strong performance.
>
>     *Table 4: Effect of the task-graph threshold τ on cluster count and relative performance.*
>
>     | **Threshold** | $\tau=0$ | $\tau=0.05$ | $\tau=0.1$ | $\tau=0.15$ | $\tau=0.2$ | $\tau=0.25$ | $\tau=0.3$ | $\tau=0.35$ | $\tau=0.45$ | w/o Cluster |
>     | --- | --- | --- | --- | --- | --- | --- | --- | --- | --- | --- |
>     | **# Cluster** | 3 | 3 | 3 | 3 | 3 | 3 | 4 | 7 | 12 | 1 |
>     | **Rel. Avg.↑** | 99.7% | 99.3% | 97.1% | 99.8% | **100.0%** | 99.7% | 96.7% | 97.6% | 97.1% | 95.8% |
> 3. **On categories being “already known”.**
>
>     We acknowledge that notions like perception and cognition are familiar from benchmarks such as MME, but this does not make CADC redundant. MME has **two** manually defined categories, while CADC **empirically discovers three model-driven capabilities** with different composition; their rough alignment with existing taxonomies is precisely what one expects from a meaningful *intrinsic* decomposition. Our contribution is to show that these capabilities can be **formally defined, discovered from dynamics, and used to drive data curation**, while remaining interpretable and useful.
>
>
> ## Weakness 2
>
> In CADC, the capabilities are **not** heuristically specified: the capability structure is **derived from the model’s training dynamics**, and MMT-Bench is used only as a probe suite on which these dynamics are observed. In the **revised paper**, Table 5 further decouples CADC from any particular benchmark: when we **replace MMT-Bench with the ICONS task set** to build the task graph, CADC still stably discovers **three** capabilities and reaches **102.8%** of full-data performance, **significantly outperforming the task-driven ICONS method itself (Fig. 3(a))**.
>
> *Table 5: Cluster count and relative performance when discovering capabilities on different target datasets.*
>
> | **Method** | **Target Datasets** | **# Cluster** | **Rel. Avg.↑** |
> | --- | --- | --- | --- |
> | **Random** | - | - | 96.4% |
> | **CADC** | MMT-Bench | 3 | **107.1%** |
> | **CADC** | ICONS | 3 | 102.8% |

---

> > ### Author Response · Authors · 2025-11-26
> >
> > ## Weakness 3
> >
> > For the LLaVA-7B results in Table 1, we strictly follow prior work and do not use staged curriculum: CADC and all baselines are fine-tuned in a single stage on their selected subsets, so the gains there come purely from capability-aware data selection.
> >
> > To isolate contributions when sequencing *is* used, Table 3 reports a component ablation. The **w/o sequence** variant keeps all other CADC components but trains in one stage; it still clearly outperforms Random, while removing the **capability** component causes the largest drop. This shows that the gains come from the full pipeline rather than curriculum alone. The sequence stage mainly serves to **reduce interference between capabilities** once they are disentangled, and is part of CADC’s design rather than an extra advantage withheld from baselines.
> >
> > ## Weakness 4; Question 1
> >
> > We agree that understanding the computational overhead of CADC is crucial. In the revised version, we add Sec. 4.2.4 with a dedicated complexity analysis and report the cost of CADC’s data processing in Table 6.
> >
> > *Table 7: Performance of CADC under different snapshot counts .*
> >
> > |  | **Rel. Avg.↑** |
> > | --- | --- |
> > | Random | 96.4% |
> > | M = 1 | 106.4% |
> > | M = 4 (default) | **107.1%** |
> > - The entire curation pipeline is **fully offline**: it is run once to produce a curated subset that can be reused across multiple models, so the cost is amortized rather than paid for every training run.
> > - Table 7 further shows that using **fewer snapshots** substantially reduces trajectory-computation cost while still preserving most of the gains, demonstrating that CADC can operate under tighter compute budgets.
> > - Heuristic methods depend on **hand-designed rules**, whose design/tuning effort is seldom quantified. CADC is **model-driven**: capabilities and scores are derived automatically from learning dynamics.
> > - As shown in the new Fig. 4 for Qwen2-VL models, the combined cost of small-model selection plus 5% CADC training is substantially lower than training the large model on all data, while matching or exceeding its performance.
> >
> > Overall, Sec. 4.2.4, Tables 6–7, and Fig. 4 show that CADC is practically feasible and offers net efficiency gains over full-data training, rather than being less practical than heuristic approaches.
> >
> > ## Weakness 5
> >
> > We understand that the original presentation may appear inconsistent. The apparent differences mainly stem from two design choices: (i) tables report **normalized scores** (relative to 100% data) to make cross-method comparisons easy, and (ii) Table 1(a)/(b) reuse the benchmark suites of TIVE [1] and ICONS [2] respectively, so their benchmark sets and baselines are not identical. These are presentation and inheritance choices rather than differences in how experiments were run. In the revised version, we clarify this by (a) standardizing the column name to **“Rel. Avg.”** and stating in the caption that entries are normalized to the full-data baseline, (b) explicitly noting in the sub-captions of Table 1(a)/(b) which prior benchmark each follows, and (c) adding ↑ markers and a brief note that larger values are better. These changes do not affect any numbers, but make it easier to interpret the tables and compare methods.
> >
> > [1] Liu Z, et al. “Less is more: High-value data selection for visual instruction tuning.”*arXiv preprint arXiv:2403.09559* (2024).
> >
> > [2] Wu X, et al. “ICONS: Influence consensus for vision-language data selection.” *arXiv preprint arXiv:2501.00654* (2025).
> >
> > We hope these clarifications are helpful, and we kindly ask that you consider them when updating your final score and recommendation.

---

> ### Author Response · Authors · 2025-11-28
>
> Dear Reviewer KJpr,
>
> We sincerely appreciate the time and effort you have dedicated to reviewing our manuscript. We provided detailed responses to your comments two days ago and wanted to check if you had any further questions. We remain fully available to address any additional concerns you may have.

---

> ### Author Response · Authors · 2025-11-28
>
> We appreciate the reviewer for pointing us to *Harvesting Rich, Scalable and Transferable Multi-Modal Data for Instruction Fine-Tuning* (mmSSR), which we see as a highly relevant and complementary line of work. mmSSR shows that carefully designed, capability-aware **heuristic scoring** can be both scalable and empirically strong for multimodal instruction tuning at large data and model scales, and provides a rich set of manually specified vision–language capability dimensions together with practical scoring rules.
>
> CADC pursues a different but complementary goal. Rather than starting from a human-defined capability taxonomy and heuristic scorers, CADC is **fully model-driven**: it discovers a small set of intrinsic capabilities directly from the model’s gradient-based learning dynamics and derives selection scores from this intrinsic structure instead of hand-crafted capability rules. These intrinsic capabilities then organize both **which data to keep** and **how to schedule training across capabilities** in a unified pipeline.
>
> In this sense, mmSSR highlights the strength of **engineered, capability-aware selection** at scale, while CADC contributes an **intrinsic, dynamics-based capability decomposition** that requires no manual capability design and is shared across datasets and backbones. We view the two approaches as addressing the same high-level goal—capability-aware data utilization—from different design perspectives, and see CADC as a complementary, model-driven alternative rather than as competing with mmSSR on the same axis.

---

### Official Review · Reviewer_i7e7 · 2025-10-27

**Soundness:** 2
**Presentation:** 2
**Contribution:** 2
**Rating:** 4
**Confidence:** 4

**Summary:**

This paper introduces Capability-Attributed Data Curation, a framework for instruction tuning in vision–language models that replaces heuristic, task-specific curation with a capability-centric approach. CADC works in three stages, Discovery, Attribution and Curation. Experiments on LLaVA and SmolVLM across benchmarks show that CADC achieves full-data or superior performance using only 5% of training data, outperforming baselines such as COINCIDE, ICONS and TIVE. Ablations and transferability tests further demonstrate robustness and interpretability.

**Strengths:**

- The authors propose a novel algorithm that recasts data curation as capability-driven rather than task-driven, providing a principled foundation for model control and interpretability.

- The proposed method demonstrates cross-model and cross-dataset generalization, with small models generating reusable subsets for larger ones.

- The ablation study exhibits the necessity of each component and the benefit of balanced sequencing.

**Weaknesses:**

- In Table 1, most baseline results (e.g., TIVE, ICONS, COINCIDE) are reported using LLaVA-v1.5-7B, while CADC is evaluated on SmolVLM-256M. Since model architectures and capacities differ substantially, performance gains may partially stem from these discrepancies rather than the proposed method itself. For a fair comparison, baselines and CADC should be evaluated using the same model and training setup.

- The main experiments primarily focus on SmolVLM-256M trained on LLaVA-1.5 Mix665K, which is relatively small-scale. Results on larger and more diverse models (e.g., LLaVA-v1.5-7B, SmolVLM-2.2B, Qwen2-VL/Qwen2.5-VL families) and datasets would strengthen the empirical evidence and demonstrate the scalability of CADC.

- The proposed framework involves computing gradient trajectories, influence estimation, and community detection, all of which appear computationally intensive. The paper lacks a clear analysis of time and memory costs compared to existing data selection baselines and to training on the full dataset, which is essential given the method’s stated goal of improving efficiency.

**Questions:**

1. Table 2 shows that CADC’s improvements diminish for larger models. Could the authors clarify why CADC is less effective in this regime? Additionally, why are only CADC-T results reported on Vision-Flan, while direct CADC results are missing?

2. The experimental design for Table 1 is difficult to interpret. Specifically:
- Why do the benchmarks differ between the 15% data and 20% data settings?
- Why are the baselines not consistent across these two settings?
- Why is the 20% CADC result shown only in Table 1(b) but not in Table 1(a)?
Clarifying these choices would help assess the comparability of results.

---

> ### Author Response · Authors · 2025-11-26
>
> We thank Reviewer i7e7 for their careful review and helpful feedback on our experimental setup and analysis. Based on the valuable comments from all reviewers, we have made three major updates and prepared detailed clarifications to strengthen our work.
>
> **Summary of Major Updates**
>
> - **Capability robustness.** We added analyses to test the stability of capability discovery, including sweeping the task-graph connectivity threshold and replacing MMT-Bench with ICONS as the target suite. Across these variants, CADC consistently discovers three capabilities and maintains strong performance.
> - **Computational cost.** We introduced a dedicated complexity analysis that decomposes the offline cost of CADC and shows that the overall selection overhead is practically acceptable. We also study variants with fewer snapshots, which further reduce cost while preserving most of the performance gains.
> - **Scaling to larger backbones.** We extended our experiments to Qwen2-VL models up to 72B parameters, all trained on the same 5% subset selected once by CADC on SmolVLM-256M. The CADC-curated subset matches or exceeds the full-data baseline while requiring substantially lower training time for the largest model.
>
> We address the reviewer's specific points below.
>
> ## Weakness 1; Question 2
>
> We agree that a fair setup for Table 1 is essential, and we clarify the roles of the data selection model, training model, and benchmark choices below.
>
> 1. **Clarifying the data selection model and training setup.**
>     - As detailed in App. F, CADC involves two models: a **data selection model** used for data curation, and a **training model** on which we train and evaluate. CADC is explicitly designed to allow these two to differ. In Table 1, **all reported values are evaluated on LLaVA-v1.5-7B as the training model**; the name in parentheses indicates the data selection model for methods that require one. Unlike prior methods, CADC uses a much smaller and heterogeneous SmolVLM-256M for selection, while training and evaluation are still done on LLaVA-v1.5-7B, so CADC and the baselines are compared under the **same training model and evaluation protocol**.
>     - This setting is **conservative with respect to CADC**: using the same architecture for selection and training tends to reduce the selector–learner gap and benefit baselines, and using a larger selection model can more accurately capture learning dynamics [1]. Despite this, CADC with a smaller, different selector still outperforms existing methods, showing that a cheap small model can be used for data curation while enabling **more efficient training of larger models**. The CADC-T results on larger SmolVLM variants in Table 2 and the Qwen2-VL [2] experiments newly added in Fig. 4 both corroborate this cross-model transfer.
>     - Fig. 3 (a) reports results where both selection and training use the **same** model and training setup; CADC again performs best in this fully aligned setting, further indicating that the gains are not an artifact of model mismatch.
> 2. **Clarifying Table 1(a)/(b) benchmarks and data sample ratios.**
>     - Table 1(a) and 1(b) reuse comparative results from **different prior works** (TIVE [3] and ICONS [1], respectively), each with its own benchmark suite and baselines; consequently, the benchmarks and baselines in the two sub-tables are not identical. In the revised version, we add an explicit note to the caption of Table 1 to improve readability.
>     - In Table 1(a), the baselines are reported only at **15%** data. We therefore initially showed CADC at **5% and 15%**. We did not include 20% to keep the comparison fair. In the revised paper, we additionally report **20% CADC** results; CADC remains strong at this higher data fraction as well.
>
> [1] Wu X, et al. “ICONS: Influence consensus for vision-language data selection.” *arXiv preprint arXiv:2501.00654* (2025).
>
> [2] Wang P, et al. “Qwen2-vl: Enhancing vision-language model's perception of the world at any resolution.” *arXiv preprint arXiv:2409.12191* (2024).
>
> [3] Liu Z, et al. “Less is more: High-value data selection for visual instruction tuning.”*arXiv preprint arXiv:2403.09559* (2024).

---

> ### Author Response · Authors · 2025-11-26
>
> ## Weakness 2; Question 1
>
> We thank the reviewer for this comment and we summarize the evidence across architectures, model sizes, candidate datasets, and target datasets.
>
> 1. **Scalability.**
>
>     Our experiments already span multiple models and datasets: Table 1 reports results on LLaVA-v1.5-7B, Table 2 on SmolVLM-2.2B with different candidate pools (Mix665K, Vision-Flan), the revised Table 5 on different targets (MMT-Bench, ICONS), and Fig. 4 on the Qwen2-VL family (2B/7B/72B). Taken together, these results show that CADC consistently improves performance across architectures, model scales, candidate datasets, and target datasets.
>
>     *Table 5: Cluster count and relative performance when discovering capabilities on different target datasets.*
>
>     | **Method** | **Target Datasets** | **# Cluster** | **Rel. Avg.↑** |
>     | --- | --- | --- | --- |
>     | **Random** | - | - | 96.4% |
>     | **CADC** | MMT-Bench | 3 | **107.1%** |
>     | **CADC** | ICONS | 3 | 102.8% |
> 2. **Interpreting Table 2 and the “diminishing” gains.**
> In Table 2, model size is not the primary variable: the goal is to test CADC/CADC-T across different candidate pools (Mix665K vs. Vision-Flan) and to confirm they remain effective when SmolVLM is enlarged from 256M to 2.2B. Across all backbones, CADC and CADC-T still clearly outperform Random, and when the backbone struggles more in the 5% regime (so Random is very low) the absolute gap to Random is actually larger. The smaller-looking relative ratios simply reflect that all 5%-data scores for these SmolVLM models lie far below their 100%-data reference, compressing the normalized values; this is a property of the backbone’s low-data behavior, not a failure of CADC. For large VL models, our new Qwen2-VL results (Fig. 4) show that 2B/7B/72B backbones trained on the same 5% CADC subset consistently match or exceed full-data training, so CADC’s advantage does not fade as model size grows.
> 3. **Clarifying the Vision-Flan setup and CADC/CADC-T results.**
>
>     On Vision-Flan, our aim was to show scalability across candidate datasets and the offline nature of CADC. CADC-T uses a small selector (SmolVLM-256M) to curate a subset that transfers across backbones, so its one-time cost can be amortized; hence it was our primary configuration there. In the revised version, we also report full CADC results on Vision-Flan in Table 2, which align with the Mix665K trends and further confirm robustness across candidate datasets.
>
>
> ## Weakness 3
>
> We agree that a clear analysis of computational and memory costs is important for an efficiency-oriented method, and in the revised version we add Sec. 4.2.4 with a dedicated complexity analysis.
>
> 1. **Complexity breakdown and dominant cost:** In the revised version, we add Sec. 4.2.4 with a dedicated complexity analysis, and report time/memory costs in Table 6. The results show that the **dominant cost** is computing gradient trajectories; the subsequent **influence estimation and community detection steps are negligible** in both time and memory compared to gradient collection. Moreover, CADC’s data curation pipeline is **fully offline** and scalable: once the trajectories are computed and a curated subset is produced, it can be reused for multiple training runs and backbones, so the one-time cost is amortized.
>
> 2. **Reducing snapshot cost without large performance loss:** Table 7 further shows that CADC is robust to using **fewer snapshots—**reducing the number of snapshots substantially lowers the cost of gradient trajectory computation, while the corresponding performance drop is small and remains within an acceptable range. This makes CADC easier to deploy under tighter resource constraints.
>
>     *Table 7: Performance of CADC under different snapshot counts .*
>
>     |  | **Rel. Avg.↑** |
>     | --- | --- |
>     | Random | 96.4% |
>     | M = 1 | 106.4% |
>     | M = 4 (default) | **107.1%** |
> 3. **Overall efficiency vs. full-data training on large models: T**he newly added Fig. 4 examines CADC on large Qwen2-VL models. It shows that, when applying CADC to large models, the **total training cost (selection + curated training)** is significantly lower than training on **100%** of the data, while achieving better or comparable performance.
>
> We hope that these updates help resolve your concerns, and we would appreciate it if you could reflect them in your final assessment and recommendation.

---

> ### Author Response · Authors · 2025-11-28
>
> Dear Reviewer i7e7,
>
> We sincerely appreciate the time and effort you have dedicated to reviewing our manuscript. We provided detailed responses to your comments two days ago and wanted to check if you had any further questions. We remain fully available to address any additional concerns you may have.

---

### Official Review · Reviewer_RTQv · 2025-11-01

**Soundness:** 3
**Presentation:** 3
**Contribution:** 2
**Rating:** 4
**Confidence:** 3

**Summary:**

This paper introduces Capability-Attributed Data Curation (CADC), a framework for selecting and sequencing instruction-tuning data for vision–language models based on their intrinsic capabilities, which are discovered in an unsupervised manner from gradient-based learning dynamics. The work addresses efficiency and controllability issues in instruction tuning by clustering tasks according to how the model learns them, mapping training data to these latent capabilities via influence analysis, and designing balanced, staged curricula.

Main contributions include: (1) a method for intrinsic capability discovery, (2) an influence-based attribution technique to map data to capabilities, (3) a curriculum arrangement and sequencing strategy for balanced capability growth, and (4) empirical results showing CADC surpasses full-data baselines with only 5% of the data; LLM baselines and pruning methods are considered.

**Strengths:**

1. Introduces an interpretable capability-based framework for data curation grounded in the model’s own learning dynamics, rather than heuristic or task-based selection.
2. Demonstrates good efficiency, outperforming baselines and full-data training while using small subsets.
3. Provides detailed experimental validation, transferability studies, and ablations that clarify the contribution of each component.

**Weaknesses:**

1. The definition and interpretation of “intrinsic capabilities” may be subjective, and the clustering method may be sensitive to hyperparameters.
2. Heavy reliance on specific experimental settings may limit generalizability outside those regimes.
3. Some methodological steps, such as curriculum sequencing based on self-influence trends, could be under-motivated or require more rigorous statistical justification.
4. Lack of scalability experiments on larger LMs (7b / 72b); Why are benchmarks inconsistent in Table 1 (a) and (b)? Do these results achieve improvement or a drop?

**Questions:**

1. How sensitive is the intrinsic capability discovery process to the choice of similarity threshold τ and the community detection parameters in the Leiden algorithm?
2. What happens when the number of discovered capabilities K is misestimated—how robust is CADC to over- or under-clustering?

---

> ### Author Response · Authors · 2025-11-26
>
> We thank Reviewer RTQv for their thoughtful review and constructive comments on our method and experiments. Based on the valuable comments from all reviewers, we have made three major updates and prepared detailed clarifications to strengthen our work.
>
> **Summary of Major Updates**
>
> - **Capability robustness.** We added analyses to test the stability of capability discovery, including sweeping the task-graph connectivity threshold and replacing MMT-Bench with ICONS as the target suite. Across these variants, CADC consistently discovers three capabilities and maintains strong performance.
> - **Computational cost.** We introduced a dedicated complexity analysis that decomposes the offline cost of CADC and shows that the overall selection overhead is practically acceptable. We also study variants with fewer snapshots, which further reduce cost while preserving most of the performance gains.
> - **Scaling to larger backbones.** We extended our experiments to Qwen2-VL models up to 72B parameters, all trained on the same 5% subset selected once by CADC on SmolVLM-256M. The CADC-curated subset matches or exceeds the full-data baseline while requiring substantially lower training time for the largest model.
>
> We address the reviewer's specific points below.
>
> ## Weakness 1；Questions 1 & 2
>
> We agree that the notion of *intrinsic capabilities* must be clearly defined and empirically grounded, and we clarify both the formal definition and its robustness below.
>
> 1. **Definition and motivation of “intrinsic capabilities”.**
>     - In Sec. 2 and App. A, we formally define an intrinsic capability as “a latent skill such that performance on any task can be factorized into contributions from one or more of these capabilities.” This notion is not hand-crafted: as described in Sec. 3.1, CADC discovers capabilities directly from the model’s learning dynamics by building a task graph from gradient similarities and applying community detection. The only subjective part is the **labels** we assign to capabilities (e.g., “perceptual recognition”), which are for human interpretability only and do not affect the method; the algorithm operates purely on unlabeled capability indices.
>     - This capability view turns “what the model needs to learn” into a small set of skill axes instead of a monolithic score. Data curation is then done to cover each capability, rather than optimizing a single aggregate metric. Empirically, Fig. 1 (radar plot) shows that CADC maintains strong performance on **all** capabilities without creating new blind spots, and Table 1 / Fig. 3(a) demonstrate that this leads to higher data efficiency at the same data budget.
> 2. **Sensitivity to $\tau$, Leiden parameters, and misestimation of $K$.**
>     - Regarding robustness, we use a standard Leiden implementation with default settings [1,2,3]; the only hyperparameter is the similarity threshold $\tau$ for graph connectivity. In the revised version (Table 4), we sweep $\tau$ and report both the resulting number of clusters $K$ and performance. For a wide range $\tau \le 0.25$, the discovered $K$ is stably 3 and the relative average performance stays high (96.3%–100.0%). When $\tau$ is increased and $K$ grows (e.g., 4–12), CADC still outperforms both **w/o Cluster** ($K = 1$) and the human-label grouping (capability ablation of Table 3). Across all $\tau$ we try, Leiden never collapses to $K < 3$. These results indicate that capability discovery in CADC is not sensitive to fine-tuning of $\tau$ or $K$ and remains beneficial over a broad range of settings.
>
> *Table 4: Effect of the task-graph threshold τ on cluster count and relative performance.*
>
> | **Threshold** | $\tau=0$ | $\tau=0.05$ | $\tau=0.1$ | $\tau=0.15$ | $\tau=0.2$ | $\tau=0.25$ | $\tau=0.3$ | $\tau=0.35$ | $\tau=0.45$ | w/o Cluster |
> | --- | --- | --- | --- | --- | --- | --- | --- | --- | --- | --- |
> | **# Cluster** | 3 | 3 | 3 | 3 | 3 | 3 | 4 | 7 | 12 | 1 |
> | **Rel. Avg.↑** | 99.7% | 99.3% | 97.1% | 99.8% | **100.0%** | 99.7% | 96.7% | 97.6% | 97.1% | 95.8% |
>
> [1] Ran Y, et al. “Machine learning informed by micro-and mesoscopic statistical physics methods for community detection.” CHAOS, 2025.
>
> [2] Gilad G, et al. “From Leiden to Tel-Aviv University (TAU): exploring clustering solutions via a genetic algorithm.” PNAS nexus, 2023.
>
> [3] Moriano Salazar P, et al. “On the Robustness of Network Community Structure Under Addition of Edges.” ORNL, 2020.

---

> > ### Author Response · Authors · 2025-11-26
> >
> > ## Weaknesses 2 & 4
> >
> > We thank the reviewer for the opportunity to clarify the generality and scalability of CADC, as well as the design of Table 1.
> >
> > 1. **Generality across architectures, model sizes, candidate data, and targets:** Our experiments are not restricted to a single architecture or data regime. Table 1 and Fig. 3(a) show that CADC improves performance across **two distinct VL architectures** (LLaVA and SmolVLM). Table 2 further demonstrates **scaling within SmolVLM** as we increase model size, and also evaluates CADC under **different candidate data pools** (Mix665K and Vision-Flan), indicating that the method is not tied to a particular source of training data. In the revised version, we additionally include Table 5, which evaluates CADC under **different target datasets (MMT-Bench and ICONS)**, showing that the gains are not specific to a single target benchmark. Beyond medium-sized models, Table 1 already includes results on **LLaVA-7B**, and in the revised version we add Fig. 4, which evaluates CADC on **Qwen2-VL [1] at 2B, 7B, and 72B**, demonstrating consistent benefits as we scale to stronger backbones.
> >
> >     *Table 5: Cluster count and relative performance when discovering capabilities on different target datasets.*
> >
> >     | **Method** | **Target Datasets** | **# Cluster** | **Rel. Avg.↑** |
> >     | --- | --- | --- | --- |
> >     | **Random** | - | - | 96.4% |
> >     | **CADC** | MMT-Bench | 3 | **107.1%** |
> >     | **CADC** | ICONS | 3 | 102.8% |
> > 2. **Clarification on Table 1(a)/(b) benchmarks:** We utilized the exact numbers reported in the baselines' original papers to ensure we were not disadvantaging them by re-running them under potentially suboptimal conditions. Because those source works adopt different benchmarks, the benchmarks appearing in Table 1(a) and 1(b) are not identical. In the revised version, we explicitly state this in the captions of the sub-tables to avoid confusion.
> > 3. **Interpretation of relative scores:** In both sub-tables, each entry is that method’s score divided by its own 100%-data score, so values above 100 indicate an improvement over full-data training (and larger is better). In the revised version, we add arrow markers to make this interpretation explicit.
> >
> > [1] Wang P, et al. “Qwen2-vl: Enhancing vision-language model's perception of the world at any resolution.” *arXiv preprint arXiv:2409.12191* (2024).
> >
> > ## Weakness 3
> >
> > We agree that the curriculum based on self-influence trends should be well-motivated and statistically supported, and we clarify both the motivation and the empirical evidence here.
> >
> > 1. **Motivation:** Prior work such as DMT [1] has shown that staging training on different data regimes can be beneficial, and TIVE [2] has used self-influence as proxies for learning difficulty. We combine these ideas at the level of *intrinsic capabilities*: we use the self-influence trajectories of tasks within each capability to estimate how quickly that capability is learned, and then schedule training stages from “easier / faster-to-learn” capabilities to “harder / slower-to-learn” ones. Staging over capabilities also helps reduce mutual interference between them, in line with the staged-training intuition in DMT.
> > 2. **Empirical and statistical evidence:** Table 3 reports a full ablation over all permutations of capability orders. The order we use in CADC, $(c_{1} \prec c_{2} \prec c_{3})$, is consistent with the self-influence trends shown in Fig. 5 (right): the capability trained first exhibits the fastest early growth, and later stages focus on capabilities whose self-influence grows more slowly. In contrast, the order $(c_{2} \prec c_{3} \prec c_{1})$ contradicts these trends: the first stage is devoted to $c_{2}$, which has the slowest initial self-influence growth; the second stage then switches to $c_{3}$, which at that point has the slowest growth among all capabilities; and by the time the third stage trains $c_{1}$, its self-influence growth advantage over the others is no longer as pronounced as in earlier stages. This sequence repeatedly allocates full stages to capabilities with the smallest marginal gain and accordingly yields the worst performance among all permutations in Table 3. Taken together, Table 3 and Fig. 5 provide both a positive example (our default sequence) and a negative counterexample, indicating that the self-influence–guided curriculum is grounded in observed learning dynamics rather than being ad hoc.
> >
> > [1] Dong G, et al. “How abilities in large language models are affected by supervised fine-tuning data composition.” ACL, 2024.
> >
> > [2] Liu Z, et al. “Less is more: High-value data selection for visual instruction tuning.”*arXiv preprint arXiv:2403.09559* (2024).
> >
> > We appreciate the concerns you raised. If the new analyses and experiments alleviate these major concerns, we would be very grateful if you could reconsider your overall evaluation in the final score and recommendation.

---

> ### Author Response · Authors · 2025-11-28
>
> Dear Reviewer RTQv,
>
> We sincerely appreciate the time and effort you have dedicated to reviewing our manuscript. We provided detailed responses to your comments two days ago and wanted to check if you had any further questions. We remain fully available to address any additional concerns you may have.

---

### Author Response · Authors · 2025-11-29

### 1. Recognized Strengths Across Reviewers

Across reviews, there is clear agreement on several positive aspects of our work. Reviewers **RTQv**, **i7e7**, and **xEJf** all recognize that CADC offers a *novel, capability-centric and interpretable* framework for instruction-tuning data curation, grounded in gradient-based learning dynamics rather than ad-hoc, task-driven heuristics. They highlight CADC’s **data efficiency**: using as little as 5% of the instruction-tuning data, CADC can match or surpass full-data training while outperforming recent data-selection baselines such as TIVE, ICONS, and COINCIDE. Reviewers **RTQv**, **i7e7**, and **KJpr** further commend the *completeness of the pipeline and empirical study*, noting that CADC covers the full process from capability discovery to data attribution and curriculum design, and is supported by ablations, transferability studies, and cross-model / cross-dataset experiments that clarify the contribution of each component. Even the more critical reviews explicitly regard CADC as a promising, principled direction for controllable instruction tuning in multimodal models.

### 2. Core Contribution of CADC

Conceptually, our core contribution is to reframe data curation for multimodal instruction tuning around *intrinsic capabilities* rather than externally defined tasks or scalar scores. CADC (i) **discovers** a small set of intrinsic capabilities by clustering task-level learning trajectories on a probe benchmark, turning the model’s training dynamics into interpretable “skill axes”; (ii) **attributes** training examples to these capabilities via trajectory-based influence estimation, forming capability-specific data pools; and (iii) **curates** capability-aware subsets by jointly deciding budget allocation and sequencing across capabilities. This shifts instruction tuning from selecting items for a black-box metric to *designing curricula over latent skills*. Empirically, this capability-centric view yields state-of-the-art data efficiency and robust, balanced performance across benchmarks and model families, and even enables a small “selector” model to curate reusable subsets for larger VLMs.

### 3. Revisions Addressing Reviewers’ Main Concerns

In the rebuttal and revised draft, we strengthen the paper precisely along the main lines raised in the reviews, while preserving the overall capability-centric vision of CADC.

- To address concerns about the **definition, novelty, and robustness** of intrinsic capabilities (RTQv, KJpr, xEJf), we add a formal definition, clarify the distinction from prior “concepts/skills” (e.g., in COINCIDE), and provide robustness studies that sweep the task-graph threshold and even **replace MMT-Bench with ICONS** as the target suite, consistently recovering three stable capabilities and maintaining strong gains.
- To address questions about **scalability and fairness of comparisons** (i7e7, xEJf), we clarify the separation between the *selection model* and *training model*, ensure that all main comparisons use the same training backbone (LLaVA-v1.5-7B), and extend experiments to **Qwen2-VL models up to 72B parameters**, all trained on a single 5% CADC subset selected once by a small SmolVLM-256M selector, where they match or exceed full-data baselines.
- In response to concerns about **computational cost and practicality** (i7e7, KJpr, xEJf), we introduce a dedicated complexity analysis and snapshot ablation: we show that the one-time offline cost is dominated by gradient-trajectory collection on a small model, that influence estimation and clustering are negligible, and that variants with fewer snapshots retain most of CADC’s gains while further reducing cost.

Together, these updates reinforce that CADC is not only conceptually principled and empirically strong, but also robust, scalable, and practically feasible as a capability-aware paradigm for instruction-tuning data curation.

---

### Meta-Review · Area_Chair_wYHt · 2026-01-03

**Summary:**

Across reviews, there is agreement that CADC is an interesting and potentially useful “capability-centric” framing for instruction-tuning data curation, with promising data-efficiency results. However, the recommendation leans reject because multiple reviewers raised (i) concerns that the “intrinsic capabilities” construct and the curriculum/sequencing logic are not yet justified rigorously enough, (ii) questions about robustness/generalizability beyond the specific experimental regimes, and (iii) practicality concerns (computational overhead / scalability) and some clarity issues in the experimental presentation (e.g., inconsistent tables / settings).

**Reviewer Concerns:**

Addressed: scalability to large VLMs (new Qwen2-VL results), computational overhead analysis + snapshot ablations, clarification that training/eval backbones match baselines (selector vs training model), robustness to τ / K behavior, and clearer explanation of Table 1(a)/(b) benchmark mismatch.

Still outstanding:
- Even with added definition/ablations, concerns remain about how “intrinsic capabilities” and the curriculum/sequencing decisions are theoretically/statistically motivated (vs. plausible but somewhat heuristic), and how reliably this transfers across broader settings without careful tuning.
- Practical competitiveness vs. simpler heuristic curation methods is still not fully settled for the typical user setting (end-to-end cost, reproducibility, and when the extra machinery is worth it).
- Novelty/positioning vs prior “skills/concepts” curation could be sharpened in the final write-up.
- Some presentation inconsistencies are fixable but should be cleaned carefully.

**Reviewer Scores:**

RTQv (4): likely stays at 4; rebuttal helps on τ/K robustness and scaling, but concerns about subjectivity/justification and generalizability likely persist.
i7e7 (4): likely stays at 4; rebuttal addresses fairness/scaling/cost partially, but the reviewer’s broader concerns about experimental design clarity and overall practicality would likely remain unless fully reflected in the revision.
KJpr (4): likely stays at 4; cost/practicality discussion helps, but they explicitly benchmark practicality against scalable heuristic approaches, which may remain a sticking point.
xEJf (→6): increases to 6 as stated by the reviewer.

---

### Decision · Program_Chairs · 2026-01-26

Reject